# Improved immune responses and tuberculosis protection by aerosol vaccination with recombinant BCG expressing ESX-1 from *Mycobacterium marinum*

Fadel Sayes[1]*, Wafa Frigui[1], Alexandre Pawlik[1], Cécile Tillier[1], Magali Tichit[2], David Hardy[2], Roland Brosch [1]*

**1** Institut Pasteur, Université Paris Cité, Unit for Integrated Mycobacterial Pathogenomics, CNRS UMR 6047, Paris, France, **2** Institut Pasteur, Histopathology Platform, Paris, France

* fadel.sayes@pasteur.fr (FS); roland.brosch@pasteur.fr (RB)

## Abstract

The currently licensed anti-tuberculosis (TB) vaccine, *Mycobacterium bovis* BCG, provides limited protection against pulmonary TB in adolescents and adults, the main cause of TB transmission and mortality. To obtain an improved BCG-based vaccine candidate with increased immune signalling but still low virulence, we have previously generated a recombinant BCG strain named BCG::ESX-1$^{Mmar}$, which is heterologously expressing ESX-1 functions of *Mycobacterium marinum* and thereby modulates the host innate immune responses via phagosomal rupture-associated induction of type I interferon production and enhanced inflammasome activity, leading to superior protection against TB disease in murine infection models. As protection may also vary with the route of vaccination, here, we have explored aerosol vaccination relative to subcutaneous vaccination, using BCG Pasteur and BCG::ESX-1$^{Mmar}$. We found that mice vaccinated via the aerosol route with BCG Pasteur or BCG::ESX-1$^{Mmar}$ both yielded higher frequencies of CD4$^+$ and CD8$^+$ Th1 activated effectors and T effector memory cells in the lungs compared to subcutaneously immunised mice, whereas comparable polyfunctional Th1 (IL-2, TNF-α and IFN-γ) cytokine-producing subsets were observed in the spleens of all vaccinated mice. Significantly higher IL-17A responses without severe lung pathology were seen in the lungs of aerosol-vaccinated mice associated to local and transient inflammatory cytokine responses and immune cell infiltrations. In contrast to the subcutaneous route, aerosol vaccination elicited high amounts of humoral IgG and IgA responses in the bronchoalveolar lavage fluid and induced a substantial number of lung CD4$^+$ and CD8$^+$ T cells expressing CD69$^+$ CD103$^+$ tissue-residency markers. These effects led to significant improved protection against *M. tuberculosis* and reduced lung pathology in aerosol-vaccinated mice compared to subcutaneously vaccinated mice. Moreover, BCG::ESX-1$^{Mmar}$ vaccine induced enhanced T-cell immunity and superior protection

**Data availability statement:** All relevant data are within the manuscript and its Supporting information files.

**Funding:** This project was partly supported by the Agence Nationale pour la Recherche (grant ANR-10-LABX62-IBEID to R.B.) and the TBVAC-HORIZON project, (European Union's HORIZON program under Grant No. 101080309 to R.B.). The funders had no role in study design, data collection and analysis, decision to publish, or preparation of the manuscript.

**Competing interests:** The authors have declared that no competing interests exist.

compared to parental BCG Pasteur for both vaccination routes and thereby represents an interesting candidate for developing improved vaccination strategies against TB.

## Author summary

Anti-tuberculosis vaccine efficacy is influenced by multiple parameters, including the immunogenicity of the vaccine strain, the type of preclinical host model used, and the route of vaccination. Given recent advances in the field of mucosal vaccination, in the current study we were particularly interested to explore and compare aerosol-based vaccination with standard subcutaneous vaccination in a C57BL/6J mouse model using our recently developed recombinant BCG::ESX-1$^{Mmar}$ vaccine candidate in comparison with parental BCG Pasteur. Our results show that in this setting the protective efficacy of mucosal vaccination was superior to subcutaneous vaccination for both vaccine strains, whereby the use of BCG::ESX-1$^{Mmar}$ induced additional benefits in terms of bacterial load reduction compared to standard BCG Pasteur. Taken together, we propose that aerosol vaccination using BCG::ESX-1$^{Mmar}$ as live-attenuated vaccine candidate is a promising and powerful combination for obtaining improved protection against an *M. tuberculosis* challenge, a concept that can now be tested in other animal models in a perspective of a putative clinical trial.

## Introduction

*Mycobacterium tuberculosis*, the principal causative agent of human tuberculosis (TB), remains a major global health threat [1] and improved strategies are needed to better control the global TB situation. However, the only currently licensed anti-TB vaccine used at a large scale to vaccinate newborns and infants, mainly in countries with high TB incidence, is represented by the Bacille Calmette and Guerin (BCG) vaccine, an attenuated variant of *Mycobacterium bovis* which has been generated more than 100 years ago [2,3]. Despite the good protection conferred against severe, disseminated forms of TB in children, the very wide use of BCG has not been able to stop the global TB epidemic, which is driven by pulmonary TB cases in adolescents and adults that may occur despite BCG vaccination [4,5]. Among the various factors that could be involved in the limited protection provided by BCG vaccination against pulmonary TB, the absence or solely partial secretion of certain key mycobacterial antigens, as well as the rather weak ability of BCG vaccines to induce long-lasting CD8$^+$ T memory responses, might play a role [6,7]. From previous work it is known that BCG lacks a 9.5 kb-sized genomic region, called Region of Difference 1 (RD1) [8,9], which in tubercle bacilli is encoding the ESX-1 type VII secretion system involved in the secretion of key proteins for host-pathogen interaction [10,11]. To overcome these potential short-comings of BCG, we have previously constructed

a recombinant BCG strain, named BCG::ESX-1*Mmar* that heterologously expresses the *esx-1* region of *Mycobacterium marinum*, an aquatic mycobacterium that harbours an ESX-1 system similar to *M. tuberculosis* [7]. The introduction of this ESX-1 system into BCG has enhanced host innate immune signalling and induced broader adaptive T-cell responses against ESX-1-secreted proteins compared to parental BCG, while maintaining low virulence in immunocompromised SCID mice, a feature that is different to a related recombinant BCG strain named BCG::RD1 or BCG::ESX-1*Mtb* that expresses the ESX-1 system of *M. tuberculosis* and which shows enhanced protective efficacy combined with enhanced virulence in preclinical models [12,13]. While the molecular reasons for the lower virulence of BCG::ESX-1*Mmar* relative to BCG::RD1/BCG::ESX-1*Mtb* are currently unknown, sequence variations [7] potentially resulting in different expression and/or interaction levels of the introduced ESX-1 core systems and the BCG-encoded ESX-1 secretion-associated protein cluster EspACD might account in part for this effect.

To improve anti-TB vaccination efficacy, several possibilities show promise, including (i) development and use of more protective vaccine strains, (ii) adoption of more efficient routes of administration and (iii) employment of heterologous prime-boost vaccination protocols (priming with live-attenuated prophylactic vaccine and boosting once or more with booster vaccine before or after *M. tuberculosis* exposure/infection). Combination of these strategies may avoid the development of active TB disease and prevent late reactivation of the infection ([4,14]).

Among the various routes of administration used for vaccination with different anti-TB vaccine candidates, the mucosal route for immunisation represents a promising choice with multiple advantages over the classical intradermal or subcutaneous routes of immunisation in mouse and non-human primate (NHP) models [15–17]. Since the lungs are the primary site of *M. tuberculosis* entry and initial infection and also represent the locus for reactivation of the TB disease, the pulmonary mucosal vaccination has been a focus of interest [16]. Indeed, the mucosal immunisation mimics the natural route of *M. tuberculosis* infection and triggers a rapid expansion of T-cell immunity and other innate immune effectors in the airways which may reduce lung pathology [18] and confer improved protection against development of TB disease [17,19–21]. Besides mucosal vaccination routes, intravenous vaccination (i.v.) has generated very promising results in NHP models [22,23], although results for a selected i.v. mouse vaccination model showed no significant reduction of the mycobacterial load compared to classical, intradermal vaccination [24].

In the present study, we have thus explored aerosol vaccination of mice with BCG::ESX-1*Mmar* as well as BCG Pasteur in comparison with subcutaneous vaccination using the same two strains. Our results provide evidence of early key innate cytokine responses with higher frequencies of Th1 effectors and effector memory cells together with significantly greater humoral IgG and IgA responses in the airways of aerosol vaccinated mice compared to the subcutaneous vaccination route. Importantly, only aerosol vaccination was able to induce IL-17A responses and substantial CD4+ and CD8+ T cells expressing resident memory markers in the primary site of *M. tuberculosis* infection leading to significant improved TB protection and lower lung pathology compared to subcutaneous counterparts.

Finally, we show that the BCG::ESX-1*Mmar* vaccine candidate displayed a superior T-cell immunity and enhanced anti-mycobacterial protective efficacy in the lungs and spleens when mice were vaccinated via aerosol or subcutaneous routes compared to the parental BCG Pasteur strain. Our data suggest that an aerosol-based mucosal vaccination with BCG::ESX-1*Mmar* represents a promising concept worth to be considered for the development of next generation vaccine strategies against *M. tuberculosis*.

## Results

### Aerosol immunisation of mice with different BCG vaccine strains elicits local and transient inflammatory responses without clinical or tissue pathologies

The main objectives of this work were the evaluation and comparison of host specific immune responses, safety and protective efficacy of BCG::ESX-1*Mmar* and parental BCG Pasteur in an aerosol vaccination mouse model relative to the standard subcutaneous vaccination model. To do so, C57BL/6J female mice were either vaccinated with 5 x 10$^5$ CFU/

mouse via the subcutaneous route, or with ca. 1 x 10³ CFU/mouse via the aerosol route, as confirmed by CFU counting at day 1 post-immunisation (Fig 1A-B). As expected, neither BCG Pasteur nor BCG::ESX-1*Mmar* bacteria were detected in the organs of subcutaneously immunised mice at any time-point following vaccination, whereas the presence of the vaccine strains was highest in the lungs and spleens of aerosol-immunised mice at four weeks post-immunisation, after which numbers started to decline (Fig 1B). The mycobacterial loads were correlated with host pro-inflammatory cytokine responses detected in the airways of the vaccinated mice (Fig 1C). We observed higher IL-1β, IL-6 and TNF-α productions at four weeks post-immunisation and that the BCG::ESX-1*Mmar*-immunised mice elicited significantly higher amounts of IL-1β compared to BCG Pasteur counterparts in the early phase following aerosol vaccination (Fig 1C), which is in accordance with the observed *in vitro* ESX-1-dependent enhanced inflammasome activation [7].

Despite the local inflammation observed in the lungs of aerosol-immunised mice, practically no significant body weight loss nor any other pronounced clinical symptoms were observed at any time-point following aerosol or subcutaneous vaccination (S1A Fig). In contrast to the lungs of subcutaneously immunised mice, which displayed a normal lung histology at eight weeks following vaccination, histological evaluations of the lungs of aerosol-immunised mice showed important immune cell infiltrations in the perivascular and peribranchial spaces without obvious tissue lesions (S1B Fig). We noted qualitatively similar innate and adaptive immune cell infiltration profiles in the inflamed regions of the lungs for BCG Pasteur- and BCG::ESX-1*Mmar*-aerosol-immunised mice. Significantly higher proportions of (i) monocytes/macrophages, (ii) neutrophils and (iii) both CD4⁺ and CD8⁺ T lymphocytes were detected in the airways of aerosol-vaccinated mice compared to the subcutaneous vaccination route, as evaluated by microscopy observations, flow cytometry analyses and immunohistochemistry on lung sections (S1C and S2A Figs).

## Aerosol immunisation is highly immunogenic and elicits both humoral and cellular Th1 cell responses in the airways of vaccinated mice

We investigated the impact of the vaccination delivery route on the immunogenicity of the vaccine and the magnitude of both mycobacteria-specific humoral and cellular responses. We detected high amounts of PPD-specific IgG and IgA responses in the bronchoalveolar lavage (BAL) fluids of aerosol-vaccinated mice at eight weeks post-immunisation, whereas subcutaneously immunised mice failed to induce such humoral responses (Fig 2A).

In our model, aerosol vaccination with BCG Pasteur or BCG::ESX-1*Mmar* induced significantly higher amounts of mycobacteria-specific T-cell IFN-γ responses in the lungs compared to the subcutaneous route at eight weeks post-immunisation (S2B Fig). Furthermore, BCG::ESX-1*Mmar* vaccine triggered specific IFN-γ responses against major ESX-1-secreted virulence factors and protective immunogens ESAT-6 (EsxA) and CFP-10 (EsxB), while this was not the case for BCG Pasteur-vaccinated mice due to the absence of these antigens from parental BCG [7,25]. To delineate the effector mechanisms of vaccine-induced T-cell immunity and the impact of the vaccination route on T-cell profiles, we subjected the lung T-cell responses to a fine analysis of functional CD4⁺ and CD8⁺ Th1 cells by intracellular cytokine staining (Figs 2B-C and S3-S4). Mice vaccinated via the aerosol route displayed significantly higher percentages of both CD4⁺ and CD8⁺ Th1 cytokine-producing effectors (IL-2, TNF-α and IFN-γ) in the lungs compared to subcutaneously vaccinated groups upon *in vitro* stimulation with PPD or a mixture of ESAT-6 and CFP-10 proteins for 24 hours (Fig 2B-C).

In response to subcutaneous vaccination with BCG Pasteur or BCG::ESX-1*Mmar*, PPD-specific CD4⁺ T effectors were mainly dominated by single-positive IFN-γ-producing cells, followed by double-positive TNF-α⁺ IFN-γ⁺ and polyfunctional IL-2⁺ TNF-α⁺ IFN-γ⁺ subsets. In aerosol-immunised mice, we observed further increased proportion of terminally differentiated single-positive IFN-γ-producing CD4⁺ T cells, probably resulting from continuous mycobacterial antigen expression/presentation *in vivo* due to the persistence of live mycobacteria in the lungs of these mice (Fig 2B). On the other hand, lung CD8⁺ T effectors were mainly composed of single cytokine-producing and IL-2⁺ TNF-α⁺ subsets following subcutaneous vaccination. In contrast, more diverse lung CD8⁺ T effector profiles were triggered by the aerosol vaccination route, including higher bi- and polyfunctional subsets (Fig 2C).

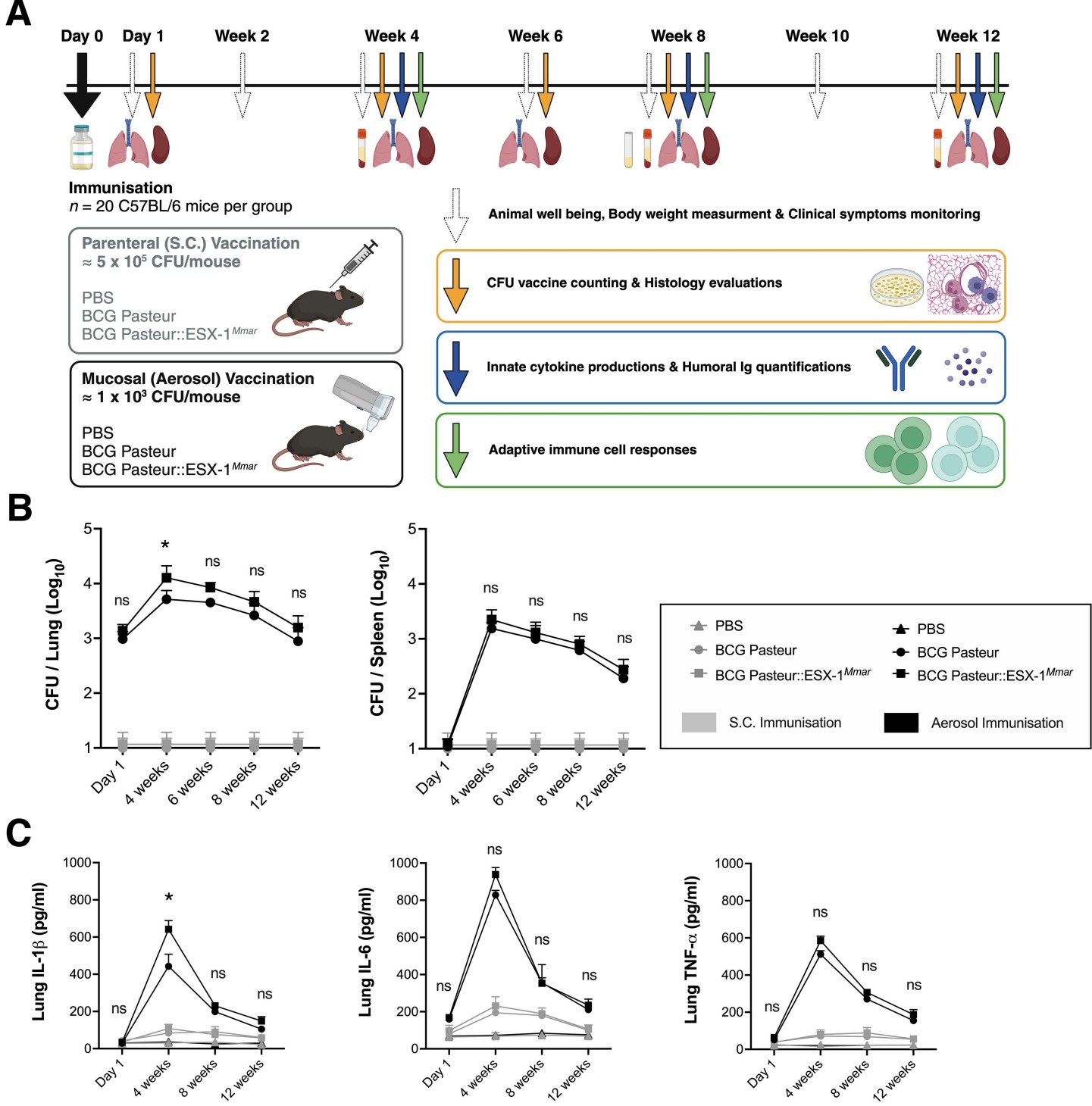

**Fig 1. Comparative evaluation model of host immune response of different BCG vaccines in mice immunised via subcutaneous or aerosol routes.** A. Protocol adopted in order to evaluate the safety, immunogenicity of the live-attenuated vaccines and their longitudinal interaction with host innate and adaptive immune systems. C57BL/6J female mice ($n=20$ per group) either left untreated or immunised subcutaneously with 5 x $10^5$ CFU/ mouse or via aerosol route with ≈ 1 x $10^3$ CFU/mouse of BCG Pasteur or BCG::ESX-1$^{Mmar}$ vaccine. At different time-points post-immunisation (solid arrows), vaccine persistence/dissemination, histology and immune responses were evaluated in the lungs, spleens and sera of mice. Scheme was

generated and licensed by BioRender under the agreement number AA294KXJSK. Created in BioRender. Sayes, F. (2026) https://BioRender.com/pmkt2gi. B. CFU counting of BCG Pasteur or BCG::ESX-1$^{Mmar}$ vaccine recovered from the lungs and spleens of vaccinated mice at different time-points post-immunisation ($n = 2$ per group, compiling data of two independent experiments). C. Kinetic of pro-inflammatory innate cytokine responses in the lungs of vaccinated mice at different time-points, as quantified by ELISA. Error bars represent SD. NS = not significant and * = statistically significant with $p < 0.05$, as determined by Unpaired $t$ test with Welch's correction. The figures were elaborated by using Prism software. The results are representative of two independent experiments. Additional related data are provided in S1-S2 Figs.

BCG::ESX-1$^{Mmar}$ vaccine induced wider Th1 effector responses compared to the BCG Pasteur parental strain regardless of the vaccination route. However, similar PPD-specific CD4$^+$ and CD8$^+$ T subset compositions were observed. Moreover, notable CD4$^+$ and CD8$^+$ Th1 effectors were detected against ESAT-6 and CFP-10 mainly in the lungs of mice vaccinated via the aerosol route (Fig 2B-C).

Interestingly, an *ex vivo* evaluation of lung CD4$^+$ and CD8$^+$ T cells without antigen stimulation revealed the presence of such Th1 cytokine-producing effectors induced by aerosol vaccination that were not detected in the lungs of mice vaccinated subcutaneously (S4A-S4B Fig) nor in unvaccinated controls stimulated with mycobacterial antigens (S4C Fig), indicating a notable vaccine-induced mucosal T cell immunity generated by aerosol immunisation. Further evaluation of the frequency and the activation/memory status of both CD4$^+$ and CD8$^+$ T effectors in the airways were addressed later in this study.

### Aerosol immunisation triggers Th17 cell responses in the lungs of vaccinated mice

Several studies have reported induction of Th17 cell responses in the airways following BCG mucosal vaccination in mice [26] and rhesus macaques [27,28], providing evidence of the critical role that IL-17A plays in the protective efficacy. In our model, we showed that only aerosol immunisation was able to elicit notable mycobacteria-specific IL-17A cytokine responses in the lungs of BCG Pasteur- and BCG::ESX-1$^{Mmar}$-vaccinated mice, as evaluated by ELISA at four weeks post-immunisation (Fig 3A). We then investigated the contribution of both lung CD4$^+$ and CD8$^+$ T cells in such specific Th17 responses by intracellular cytokine staining at a later time-point following vaccination. Indeed, we detected a significantly higher frequency of PPD-specific IL-17A-producing CD4$^+$ and CD8$^+$ T cells in the lungs of aerosol immunised mice compared to subcutaneously immunised groups at two months post-immunisation (Fig 3C). Interestingly, CD8$^+$ T lymphocytes appear to be an important source of IL-17A responses in the airways following aerosol vaccination, as judged by the percentage of specific IL-17A-producing CD8$^+$ T cells (Fig 3C) and their high cytokine expression levels measured by the mean fluorescence intensities (Fig 3D). Moreover, aerosol vaccination with BCG::ESX-1$^{Mmar}$ enhanced lung PPD-specific IL-17A$^+$ CD8$^+$ T cells compared to BCG parental strain and elicited additional IL-17A responses against major ESX-1-secreted antigens (Fig 3C).

### Aerosol immunisation triggers higher frequencies of activated effector CD4+ and CD8+ T lymphocytes, T effector memory cells and substantial T cells expressing tissue-residency markers in the lungs compared to the subcutaneous route

Induction of robust Th1 effectors and T effector memory cells subsequent to vaccination is crucial for host defence and correlates with better protection and/or control of *M. tuberculosis* infection [7,29]. In our model, we detected significant higher frequencies of activated CD4$^+$ and CD8$^+$ T lymphocytes that displayed down-regulation of CD27, CD45RB and CD62L expressions, in the lungs of aerosol-immunised mice compared to subcutaneously immunised counterparts at eight weeks post-immunisation (S5 Fig). Their frequency in the airways was lower than observed at four weeks post-immunisation (S6 Fig), suggesting a correlation between the mycobacterial burden, host inflammatory responses and recruitment/expansion of IFN-γ-producing activated T cells. In addition, a significantly higher numbers of total CD3$^+$ CD4$^+$ and CD3$^+$ CD8$^+$ T lymphocytes were recovered from the lungs of aerosol-immunised mice compared to subcutaneously immunised mice at four weeks (Fig 3B) and eight weeks (S2A Fig) post-immunisation.

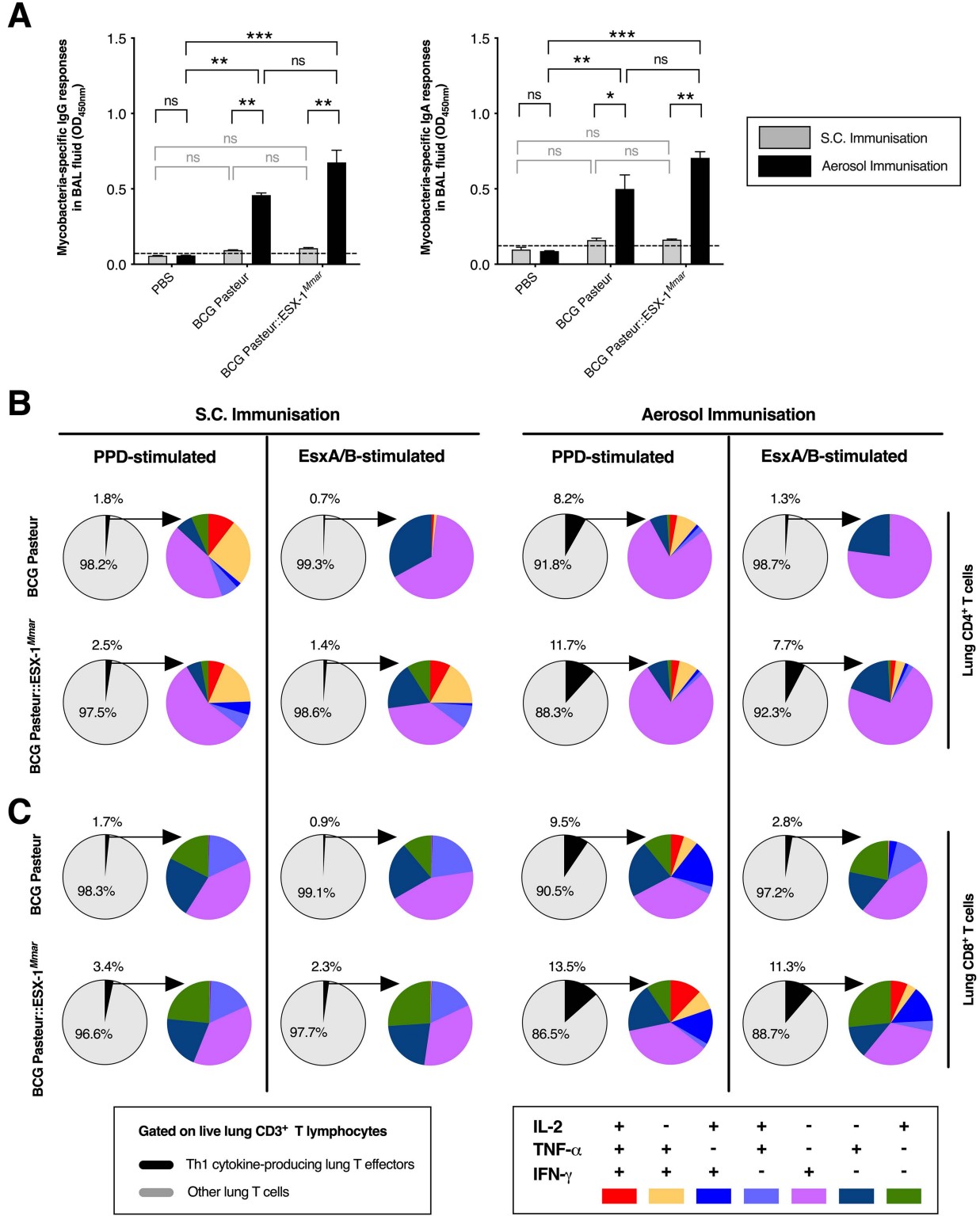

**Fig 2. Induction of humoral and Th1 cytokine-producing CD4⁺ and CD8⁺ T effectors in the lungs of mice immunised with different BCG vaccines via subcutaneous or aerosol routes.** A. Humoral immunoglobulin responses in the airways of vaccinated mice at eight weeks post-immunisation ($n = 2$ per group), as evaluated by mycobacterial-specific IgG and IgA recovered from the BAL fluid by ELISA. Error bars represent SD.

NS = not significant, *, ** and *** = statistically significant with $p < 0.05$, $p < 0.005$ and $p < 0.001$, respectively, as determined by Unpaired $t$ test with Welch's correction. B-C. Frequency of Th1 cytokine-producing CD4+ (B) and CD8+ (C) T cells within live lung CD3+ T lymphocytes in the lungs of vaccinated mice ($n = 3$ per group) at eight weeks post-immunisation subsequent to in vitro stimulation with PPD or a mixture of ESAT-6 and CFP-10 proteins for 24 hours at 37°C and 5% $CO_2$, as evaluated by intracellular cytokine staining assay to define and determine each T effector subsets producing IL-2, TNF-α and IFN-γ cytokines. At least 1,000,000 events per sample were acquired for flow cytometric analyses using FlowJo software and figures were elaborated by using Prism software. The results are representative of two independent experiments. Additional related data are provided in S2-S4 Figs.

Moreover, an increased recruitment/accumulation of both CD4+ and CD8+ T effectors expressing CXCR3 (Fig 4A) and T effector memory cells (T_{EM} defined as CD3+ CD44^{high} CD62L-) (Fig 4B) in the lungs of aerosol-immunised mice compared to subcutaneous counterparts was observed at eight weeks post-immunisation. These T cells showed high intensity of CD44 expression on both lung CD4+ and CD8+ T subsets (Fig 4C), indicating a strong vaccine-induced activation and effector/memory function of host Th1 responses subsequent to aerosol vaccination.

Previous mouse models have suggested that mucosal vaccination induces lung-associated T resident memory cells that protect against TB disease [30,31]. Hence, we evaluated the capacity of different routes of immunisation with BCG Pasteur and BCG::ESX-1^{Mmar} vaccines to induce such long-lived T effectors expressing tissue-residency markers. We found that only aerosol vaccination elicited substantial lung CD69+ CD103+ T lymphocytes, containing both CD4+ and CD8+ T cell compartments at eight weeks post-immunisation, while subcutaneous immunisation failed to induce such a response (Fig 5). These cell populations are defined as CD3+ CD45+ CD44^{high} CD62L- CD69+ CD103+ cells in mice (Fig 5A). Importantly, the BCG::ESX-1^{Mmar} vaccine generated significantly higher percentages and absolute numbers of both CD4+ and CD8+ T cell effectors expressing tissue-residency markers in the airways compared to BCG Pasteur (Fig 5B-C).

**Aerosol immunisation induces potent polyfunctional Th1 effectors in the spleens and systemic IgG responses similar to those detected by subcutaneous immunisation**

In addition to vaccine-induced humoral and T-cell immunity in the airways, we investigated the capacity of aerosol immunisation to induce Th1 effectors in the secondary lymphoid organs. This exercise showed comparable amounts of mycobacteria-specific T-cell IFN-γ responses in the spleens of both aerosol and subcutaneously immunised mice at eight weeks post-immunisation (Fig 6A). Mice vaccinated with BCG::ESX-1^{Mmar} displayed such responses also against the major ESX-1 antigens ESAT-6 and CFP-10. Moreover, we observed strong CD4+ and CD8+ T cells expressing CXCR3 and T effector memory cell responses in the spleens of these mice, although at a lower frequency in aerosol vaccinated mice relative to the ones vaccinated by the subcutaneous route (S7 Fig). In-depth characterization of the functional Th1 cell subsets and their differentiation status by intracellular staining showed that both vaccination routes with BCG Pasteur or BCG::ESX-1^{Mmar} induced globally comparable Th1 cytokine-producing CD4+ effectors with similar T subset compositions in the spleens of vaccinated mice at eight weeks post-immunisation upon in vitro stimulation with PPD (Fig 6B). The responses were dominated by terminally differentiated single-positive TNF-α+ and IFN-γ+ CD4+ T cells, followed by double-positive TNF-α+ IL-2+ CD4+ T cells. We noted that single-positive IFN-γ+ CD4+ T cells were slightly higher in the spleens of subcutaneously immunised mice while single-positive TNF-α+ CD4+ T cells were lower than in aerosol-immunised mice. Furthermore, comparable CCR6 and CXCR3 expressions on polyfunctional IL-2+ TNF-α+ IFN-γ+ cytokine-producing CD4+ T cell subsets were detected in the spleens of all vaccinated mice (Fig 6B).

In addition, we detected mycobacteria-specific cytokine-producing CD8+ T effectors in the spleens of mice that were comparable between the two vaccination routes at twelve weeks post-immunisation (S8A-B Fig). A higher percentages of TNF-α+ and IFN- γ+ bi-functional CD8+ T effectors were induced by BCG::ESX-1^{Mmar} compared to BCG Pasteur, probably due to broader responses to ESX-1 secreted antigens and/or induction of cytosolic immune signalling for the ESX-1-proficient strain compared to BCG Pasteur, leading to enhanced MHC-I-restricted epitope presentation [7,32,33]. In contrast, comparable Th1 cytokine-producing CD4+ T effectors with similar T subset compositions were induced by BCG::ESX-1^{Mmar} and BCG Pasteur in the spleen at twelve weeks post-immunisation (S8C Fig), similar to what has also

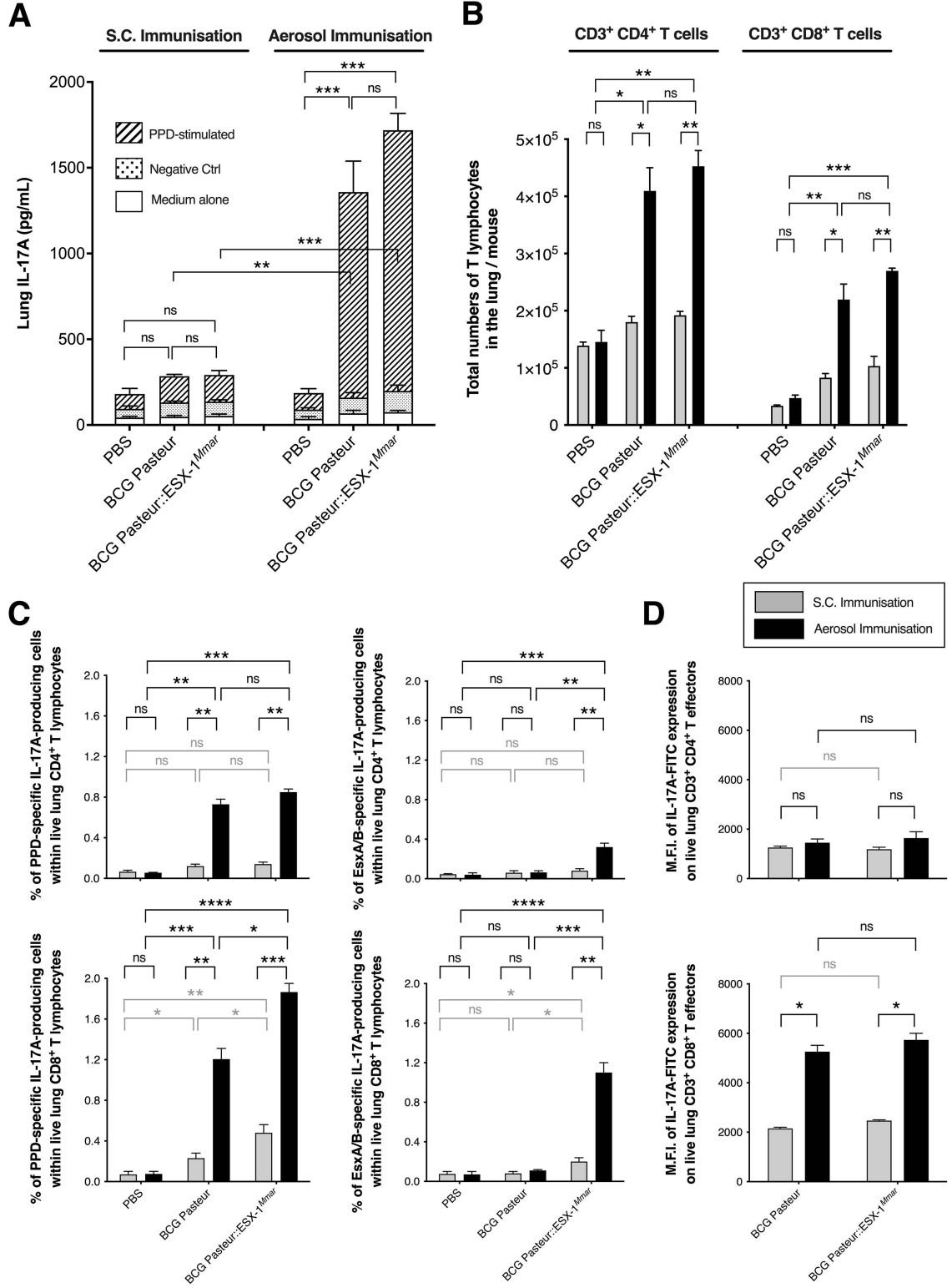

**Fig 3. IL-17A cytokine responses in the lungs of mice immunised with different BCG vaccines via subcutaneous or aerosol routes.** A. Quantification of IL-17A in the lungs of C57BL/6J mice ($n = 2$ per group) vaccinated via subcutaneous or aerosol routes as assessed at four weeks post-immunisation. Total lung cells of the immunised mice were stimulated *in vitro* with PPD during 72 hours at 37°C and 5% $CO_2$. Medium alone and MalE

protein were used as negative controls. The IL-17A amount was quantified in the culture supernatant by ELISA. B. Total numbers of CD3+ CD4+ and CD3+ CD8+ T cells recovered from the lungs of mice at four weeks post-immunisation, as determined as total numbers of cells in the Ficoll-treated lung fractions, multiplied by the percentages of CD4+ or CD8+ cells, respectively, as assessed by flow cytometry analyses. C. Frequency of IL-17A-producing CD4+ and CD8+ T cells in the lungs of vaccinated mice ($n = 3$ per group) subsequent to *in vitro* stimulation with PPD or a mixture of ESAT-6 and CFP-10 proteins for 24 hours at 37°C and 5% $CO_2$, as evaluated by intracellular cytokine staining assay on live lung CD3+ T lymphocytes. D. Mean Fluorescence Intensities (M.F.I.) of IL-17A expression on live lung CD3+ CD4+ and CD3+ CD8+ T effectors recovered from the same vaccinated mice. At least 1,000,000 events per sample were acquired for flow cytometric analyses using FlowJo software. Error bars represent SD. NS = not significant, *, **, *** and **** = statistically significant with $p < 0.05$, $p < 0.005$, $p < 0.001$ and $p < 0.0001$, respectively, as determined by Unpaired *t* test with Welch's correction. The figures were elaborated by using Prism software.

been observed at eight weeks post-immunisation. Also, the total numbers of both CD4+ and CD8+ T lymphocytes in the spleens of vaccinated mice were comparable between both vaccination routes at this time-point (S8D Fig).

As regards humoral responses, we noticed that aerosol and subcutaneous vaccinations elicited notable and long-term systemic humoral responses comparable at all time-points following vaccination, as judged by PPD-specific IgG levels quantified in the sera of vaccinated mice (Fig 6C). This finding suggests that aerosol vaccination with live-attenuated vaccines is a highly immunogenic route able to elicit not only greater mucosal (local) immune responses compared to the parenteral route, but also comparable systemic responses.

### Aerosol vaccination of mice significantly improved TB protection and showed lower lung pathology compared to parenteral vaccination route

Finally, we wondered if the enhanced lung innate immunity, the higher airway humoral responses and improved CD4+ and CD8+ Th17/Th1 activated/effector/memory responses induced by aerosol immunisation could improve protection of mice against an *M. tuberculosis* challenge (Fig 7A). The experiments showed that mice immunised with BCG Pasteur or BCG::ESX-1*Mmar* via the aerosol route displayed significant improved TB protection in the lungs compared to subcutaneously immunised groups, as judged by reduction of mycobacterial loads at four weeks post-infection (Fig 7B), as well as in terms of lower lung pathology (Fig 8). Indeed, fewer tissue lesions and cell infiltrations were found in the lungs of BCG Pasteur- and BCG::ESX-1*Mmar*- aerosol-vaccinated mice compared to subcutaneously immunised groups at four weeks post-infection. In contrast, no statistically significant differences in CFU counts were noted between aerosol and subcutaneous immunisation routes in the spleens of mice at four weeks post-infection (Fig 7C). For these mouse experiments we also noticed that vaccination with BCG::ESX-1*Mmar* was more protective than vaccination with BCG Pasteur for both immunisation routes, when using CFU counts as a readout (Fig 7B-C), as well as in depth histopathological evaluations on some key parameters such as percentages of lung damaged and consolidated areas, and inflammation seriousness scores (Fig 8B).

Together, these results obtained with our murine model highlight the potential advantages of the aerosol route of vaccination compared to the classical route of administration of live-attenuated anti-TB vaccines and also indicate that the BCG::ESX-1*Mmar* vaccine represents an improved candidate for such a revised strategy of vaccination against *M. tuberculosis*.

### Discussion

To achieve the WHO goal of enhanced control and potential eradication of TB worldwide, more effective vaccines and new anti-TB drugs are needed. An optimized vaccination strategy against TB should primarily provide a better protection of adolescents and adults, both in terms of Protection of Infection (POI) and Protection of Disease (POD), as individuals from these age-sections are often - despite BCG vaccination in their early childhood - not efficiently protected against pulmonary forms of TB disease, which represent the most efficient transmission routes for spreading *M. tuberculosis* to new hosts, keeping the TB infection cycle going. Development of improved vaccination strategies against infection and/or TB

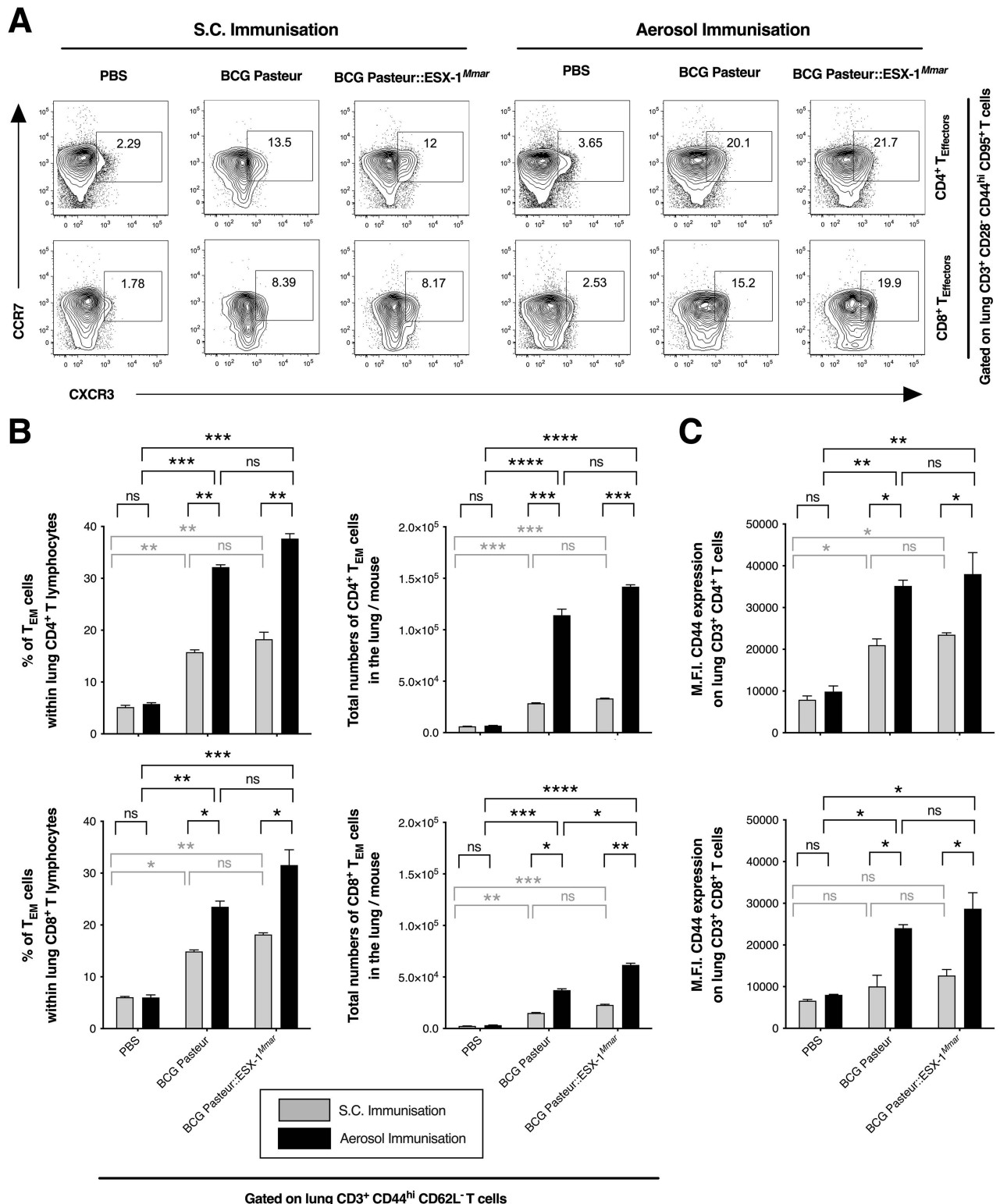

**Fig 4. Induction of T effectors and T effector memory cell responses in the lungs of mice immunised with different BCG vaccines via sub-cutaneous or aerosol routes.** A. CD4+ and CD8+ T effector cells expressing CXCR3 marker recovered from the lungs of BCG Pasteur- or BCG::ESX-1$^{Mmar}$-immunised mice via subcutaneous or aerosol routes. Cell populations were gated on CD3+ CD28- CD44$^{high}$ CD95+ T lymphocytes. Cytometric

plots represent 5% contours with outliers, representative of pool of three mice per group. B. Frequency of lung CD4+ and CD8+ T effector memory cells, defined as CD3+ CD44high CD62L- cells, among CD4+ and CD8+ T lymphocyte populations, as evaluated by flow cytometric analyses at eight weeks post-immunisation. C. Mean Fluorescence Intensities (M.F.I.) of CD44 expression on total lung CD3+ CD4+ and CD3+ CD8+ T cells recovered from the same vaccinated mice. Error bars represent SD. NS = not significant, *, **, ***, and **** = statistically significant with $p < 0.05$, $p < 0.005$, $p < 0.001$ and $p < 0.0001$, respectively, as determined by Unpaired $t$ test with Welch's correction. At least 500,000 events per sample were acquired. The obtained data were analysed using FlowJo software and figures were elaborated by using Prism software. The results are representative of two independent experiments. $T_{EM}$: T effector memory cells; Additional data are provided in S5-S6 Figs.

disease in adolescents and adults, resulting either from primo-infection, reactivation and/or re-infection, thus represents a key objective for preclinical and clinical research. One of the premises for generating improved vaccine candidates is to increase the immunogenicity relative to the standard BCG vaccines, whereby live-attenuated vaccine candidates such as VPM1002 [34,35] and/or MTBVAC [36,37] are currently evaluated in clinical trials.

In a similar attempt, we have recently generated a recombinant BCG Pasteur strain complemented with an integrating cosmid that carries the *esx-1* region of *M. marinum* [7]. The heterologous expression of the *M. marinum* ESX-1 type VII secretion system in this vaccine candidate impacted different aspects of host-pathogen interaction, including the induction of phagosomal rupture, the access to the host cytosol and enhanced immune signalling [7]. These aspects are all important parameters during mycobacterial infection, with profound influence on host innate and adaptive immune responses [7,29,32,38–41]. Indeed, the BCG::ESX-1*Mmar* strain was able to modulate the host innate immune response via phagosomal rupture-associated induction of type I interferon (IFN) responses that were transmitted through the cGAS/STING pathway, as well as enhanced inflammasome activity, resulting in higher IL-1β release and higher proportions of CD8+ T cell effectors against mycobacterial antigens and potent polyfunctional CD4+ Th1 cells specific to ESX-1-secreted antigens ESAT-6, CFP-10 and EspC [7,32]. We hypothesize that these changes are probably the main reason for the superior TB protection observed in mice vaccinated with BCG::ESX-1*Mmar* compared to BCG Pasteur and BCG Danish vaccine strains in independent murine infection models that used the H37Rv reference strain, but also hypervirulent *M. tuberculosis* strains belonging to Beijing and Haarlem strain families as challenge strains [7]. Given these encouraging previous results, we also included the BCG::ESX-1*Mmar* strain in our current study, which mainly focused on exploring different vaccination routes.

Indeed, in the current study, we compared aerosol-mediated vaccination versus standard subcutaneous vaccination in C57BL6J mice using BCG Pasteur and BCG::ESX-1*Mmar* as live-attenuated vaccine strains, evaluating their immunogenicity and space-temporal interactions with the host. As reported in detail in the results section, aerosol vaccination with BCG Pasteur and BCG::ESX-1*Mmar* was highly immunogenic, inducing strong innate and adaptive activated/effector/memory cell responses at the airway mucosal (local) level (Figs 1, 2, 3, 4, 5 and S1-S6) as well as systemic polyfunctional Th1 cytokine-producing effectors that were comparable between the two vaccination routes (Figs 6 and S7-S8).

Interestingly, only aerosol vaccination induced substantial long-lived non-circulating CD4+ and CD8+ T effectors expressing tissue-residency markers in the lung tissue of immunised mice while subcutaneous vaccination failed to do so (Fig 5). These results are in good agreement with examples from the literature, indicating that T resident memory cells play numerous immune functions and have tissue-specific and migration property-specific differentiation programs [42–44]. As such it was shown that mouse CD4+ and CD8+ T resident memory cells share immunosurveillance strategies in the lung tissue after acute viral infection and that CD4+ T resident memory reactivation plays a key role in the immune recall by triggering chemokine responses and amplifying immune cell activation [44]. Zens et al., 2016 reported that long-term protective CD4+ and virus-specific CD8+ T resident memory cells were only generated in the mouse airways subsequent to intranasal vaccination with live-attenuated influenza vaccine, similar in phenotype to those generated by influenza virus infection, but not following vaccination with injectable inactivated influenza vaccine, a finding that strongly suggests that the induction of these responses is highly dependent on the nature of vaccine and the administration route [42,45].

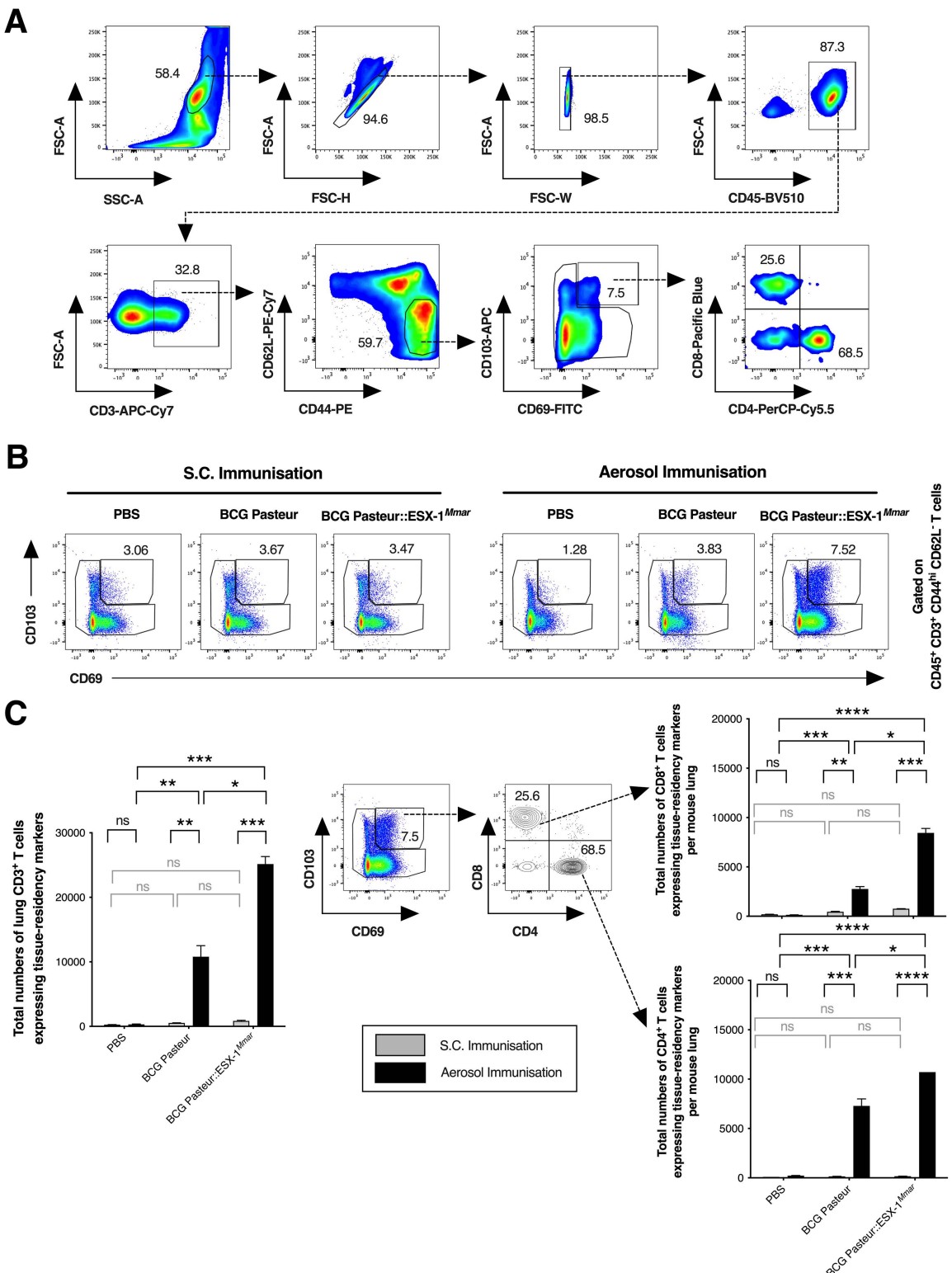

**Fig 5. Induction of T effector cells expressing tissue-residency markers in the lungs of mice immunised with different BCG vaccines via subcutaneous or aerosol routes.** A. Gating strategy and flow cytometric analyses adopted to identify both lung CD4+ and CD8+ T resident memory cell

populations. B. Frequency of CD69$^+$ CD103$^+$ T cells recovered from the lung parenchyma of mice vaccinated with BCG Pasteur or BCG::ESX-1$^{Mmar}$ via subcutaneous or aerosol routes, as evaluated by flow cytometric analyses at eight weeks post-immunisation. C. Total numbers of lung CD4$^+$ and CD8$^+$ T effector cells expressing tissue-residency markers per mouse. Error bars represent SD. NS = not significant, *, **, ***, and **** = statistically significant with $p < 0.05$, $p < 0.005$, $p < 0.001$ and $p < 0.0001$, respectively, as determined by Unpaired $t$ test with Welch's correction. At least 500,000 events per sample were acquired, representative of pool of three mice per group. The obtained data were analysed using FlowJo software and figures were elaborated by using Prism software. The results are representative of two independent experiments.

In additional to cell-mediated immunity [46], B cells and humoral antibody responses in the mucosa of vaccinated mice may efficiently contribute to host defence and the control of *M. tuberculosis* infection [47,48]. Whereas elevated IgG and IgA responses were present in the BAL fluids of aerosol-vaccinated mice, and not for subcutaneously vaccinated mice (Fig 2A), quantified serum IgG responses were strong and long-persistent for both vaccination routes (Fig 6C).

Nemeth et al., 2020 showed in a mouse model, which used a contained and persistent yet non-pathogenic infection with *M. tuberculosis*, that rapid and durable reduction of TB disease burden was achieved in these mice when they were re-exposed through an aerosol challenge with the same *M. tuberculosis* strain. This experimental model, which mimics to some extent a latent TB infection showed that pre-exposure to low dose *M. tuberculosis* can be beneficial for the host, protecting to some degree against subsequent aerosol challenge with higher doses. The observed protection is associated with elevated activation of alveolar macrophages and accelerated recruitment of *M. tuberculosis*-specific T cells to the lung parenchyma, resulting in reduced disease upon re-exposure to *M. tuberculosis* [49]. Moreover, another recent study revealed that a low-level persistent infection of mice with *Plasmodium chabaudi* generated protective immunity by impacting CD4$^+$ T cell subsets, promoting in particular both IFN-γ$^+$ and TNF$^+$ double positive and terminally differentiated IFN-γ$^+$ T effectors [50]. These results support the hypothesis of a potentially beneficial role of persistent microbial infections at a non-pathogenic level, which may create an immunologic condition that is reminiscent to the one generated by the use of live-attenuated vaccine strains.

Our study also demonstrated that mice vaccinated with BCG Pasteur or BCG::ESX-1$^{Mmar}$ via the aerosol route were significantly better protected against a challenge with virulent *M. tuberculosis* compared to mice vaccinated via the subcutaneous route, as shown by reduced lung mycobacterial loads (Fig 7B) and superior lung histology scores with lower inflammatory lesions/disease seriousness (Fig 8). Interestingly, comparable *M. tuberculosis* CFU counts were observed in the spleens of these aerosol-vaccinated mice relative to subcutaneously vaccinated mice, even with an administrated vaccine dose that was ≈ 500-fold lower in the aerosol group. These findings suggest that similar levels of CD4$^+$ and CD8$^+$ T effectors and effector memory cell responses were induced in the spleens by both vaccination routes despite different doses (Figs 6 and S7-S8), leading to a comparable protection level in this organ (Fig 7C). Concerning the lungs, we hypothesize that the superior protection seen in aerosol-vaccinated mice relative to subcutaneously vaccinated mice is likely due to an enhanced quantity and quality of sustainable Th1 and Th17 cellular immunity in this site, involving activated/effector/memory and tissue-resident T cells, in addition to humoral IgA and IgG responses.

An increased protection conferred by mucosal vaccination relative to subcutaneous vaccination was also observed in several previous studies using murine infection models. For example, it has been shown that intranasal administration of BCG in mice induced superior protection in the lungs at early time-points as well as long elevated protective splenic responses with higher frequencies of CD4$^+$ and CD8$^+$ T cells expressing IFN-γ [51]. A greater protective efficacy following intranasal BCG administration was also observed in TB-susceptible DBA/2 mice, with higher *M. tuberculosis*-specific Th1 and Th17 immune responses, as well as IgA concentration in lungs compared to subcutaneous vaccination [26]. Aguilo et al., 2014 showed that intratracheal vaccination of C57BL/6 mice with BCG confers dose-dependent superior TB protection compared to that observed by subcutaneous administration [52]. The mucosal (both intranasal and intratracheal) BCG vaccination in mice also generated T effector memory and T resident memory cells in the lung [30] and was able to induce potent lung tissue resident PD-1$^+$ KLRG1$^-$ CD4$^+$ T cells [53] that confer enhanced pulmonary protection against *M.*

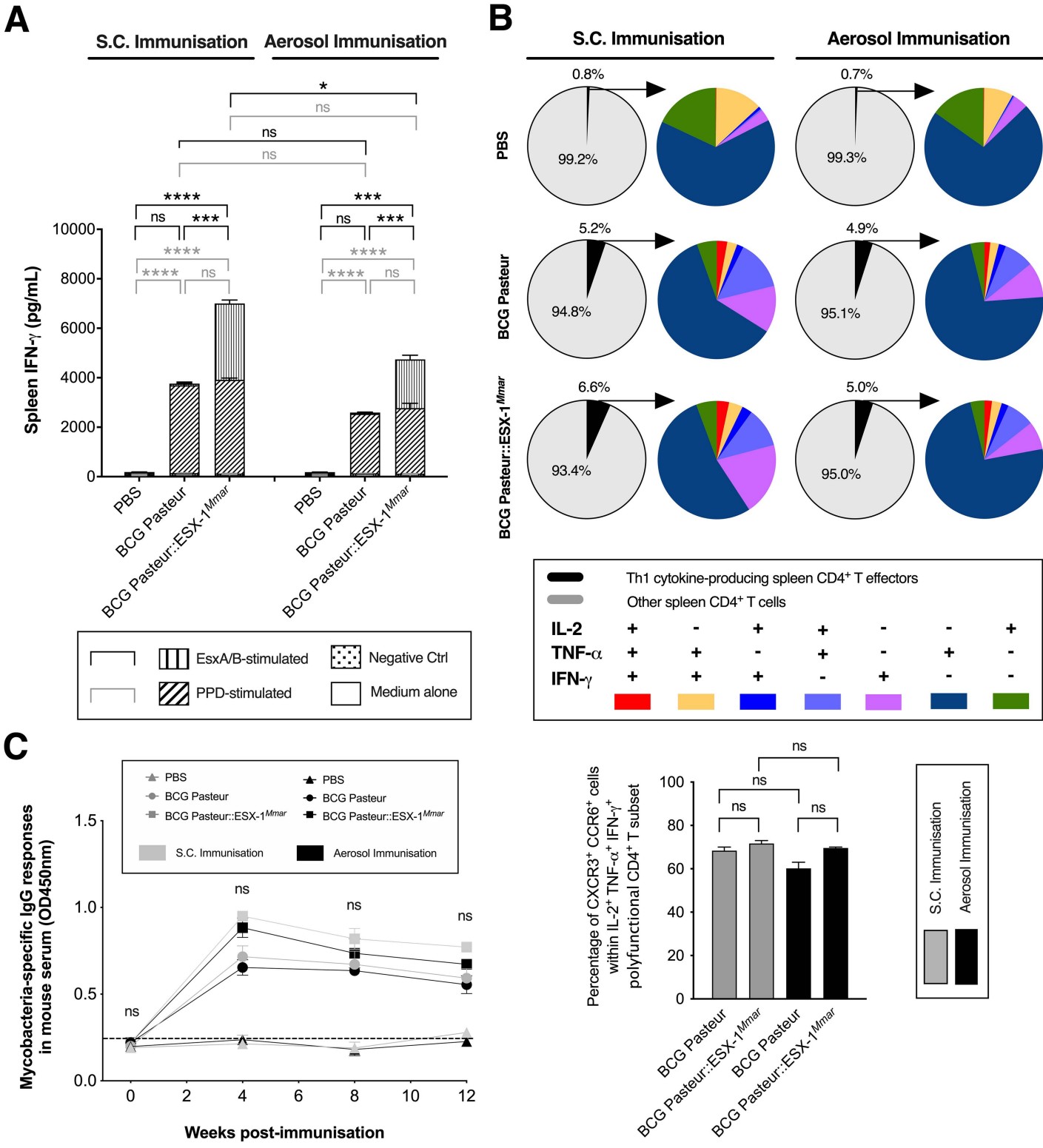

**Fig 6. Induction of polyfunctional Th1 cell responses in the spleens and humoral IgG responses in the sera of mice vaccinated with different BCG vaccines via subcutaneous or aerosol routes.** A. T-cell IFN-γ responses in the spleens of BCG Pasteur or BCG::ESX-1$^{Mmar}$-vaccinated mice via subcutaneous or aerosol routes, as assessed at eight weeks post-immunisation. Pool of total splenocytes ($n = 2$ per group) were stimulated *in vitro* with

with PPD or a mixture of ESAT-6 and CFP-10 proteins during 72 hours at 37°C and 5% $CO_2$. Medium alone and MalE protein were used as negative controls.The IFN-γ was quantified in the culture supernatant by ELISA. B. Frequency of Th1 cytokine-producing CD4+ T subsets in the spleens of vaccinated mice at eight weeks post-immunisation. Total splenocytes of vaccinated mice ($n$ = 2 per group) were stimulated in vitro with 10 μg/ml of PPD for 24 hours prior to intracellular cytokine staining (IL-2, TNF-α and IFN-γ) and flow cytometric analyses. At least 1,000,000 events per sample were acquired for flow cytometric analyses. The obtained data were analysed using FlowJo software. The results are representative of two independent experiments. C. Mycobacteria-specific IgG responses in the sera of vaccinated mice at different time-points post-immunisation ($n$ = 2 per group), as evaluated by ELISA. Error bars represent SD. NS = not significant, *, *** and **** = statistically significant with $p < 0.05$, $p < 0.001$ and $p < 0.0001$, respectively, as determined by Unpaired $t$ test with Welch's correction. The figures were elaborated by using Prism software. The results are representative of two independent experiments. Additional data are provided in S7-S8 Figs.

tuberculosis. In addition, Mata et al., 2021, showed that pulmonary but not subcutaneous administration of BCG induces lung-resident macrophage activation in mice, even after vaccine clearance [54]. It was also reported that this activation of alveolar macrophages was mainly mediated by CD4+ T cells, providing not only a long-term TB protection but also conferring heterologous protection against Streptococcus pneumoniae infection, suggesting that BCG mucosal vaccination drives potent in vivo trained innate memory-like responses. Such an additional protective effect of trained immunity might also be induced by mucosal vaccination with BCG Pasteur and BCG::ESX-1$^{Mmar}$ vaccine, and might contribute to the improved protection observed in our model. Indeed, immunohistochemistry analyses revealed a significant higher number of F4/80+ activated monocytes/macrophages in the lung parenchyma of aerosol-vaccinated mice compared to subcutaneously vaccinated mice (S1C Fig).

In conclusion, our experiments in mice revealed that both the vaccine route as well as the vaccine type influenced the degree of protection from a challenge with M. tuberculosis. Benefits were seen for aerosol vaccination compared to subcutaneous vaccination in terms of CFU reduction and histological preservation of the lung tissue. Moreover, superior sustained Th1 and Th17 CD4+ and CD8+ T effectors as well as lymphocytes expressing tissue-residency markers were induced by the vaccination with BCG::ESX-1$^{Mmar}$ compared to the BCG Pasteur parental strain, as observed at the time of challenge with M. tuberculosis (Figs 2, 3, 4, 5). This finding suggests that T-cell immunity mediated by ESX-1-secreted antigens efficiently contributes to enhanced TB protection in the lungs of mice both tested vaccination routes (Figs 7, 8). As such, our study confirms that BCG::ESX-1$^{Mmar}$ does represent an interesting new vaccine candidate and further suggests that it is a promising asset for conventional, subcutaneous as well as alternative, aerosol-based vaccination, worth to be considered for further preclinical and eventual clinical development.

## Experimental procedures (materials and methods)

### Animal studies and ethical statement

All animal-involving studies were conducted in agreement with the European and French guidelines (EC Directive 2010/63/UE and French Law 2013–118 issued on 1 February 2013). These experiments were approved by the Institut Pasteur safety committee (protocol 11.245) and by the relevant Ethics Committee (Comité d'éthique en experimentation animale 89) and by the French Ministry for Higher Education and Research (dap 180023, APAFIS #15409–2018060717283847 v1 and dap220021, APAFIS #37011–2022042811231522 v1), respectively.

### Mycobacterial strains

M. bovis BCG Pasteur was grown in Dubos broth medium (Becton Dickinson) supplemented with Albumin, Dextrose and Catalase (ADC, Becton Dickinson) at 37ºC and BCG::ESX-1$^{Mmar}$ was grown in the same medium to which hygromycin 50 μg/ml was added. Mycobacterial concentrations were determined by $OD_{600nm}$ measurement and CFU counting on Middlebrook 7H11 solid Agar medium (Becton Dickinson) complemented with Oleic acid, Albumin, Dextrose and Catalase (OADC, Becton Dickinson) after 3–4 weeks of culture at 37°C.

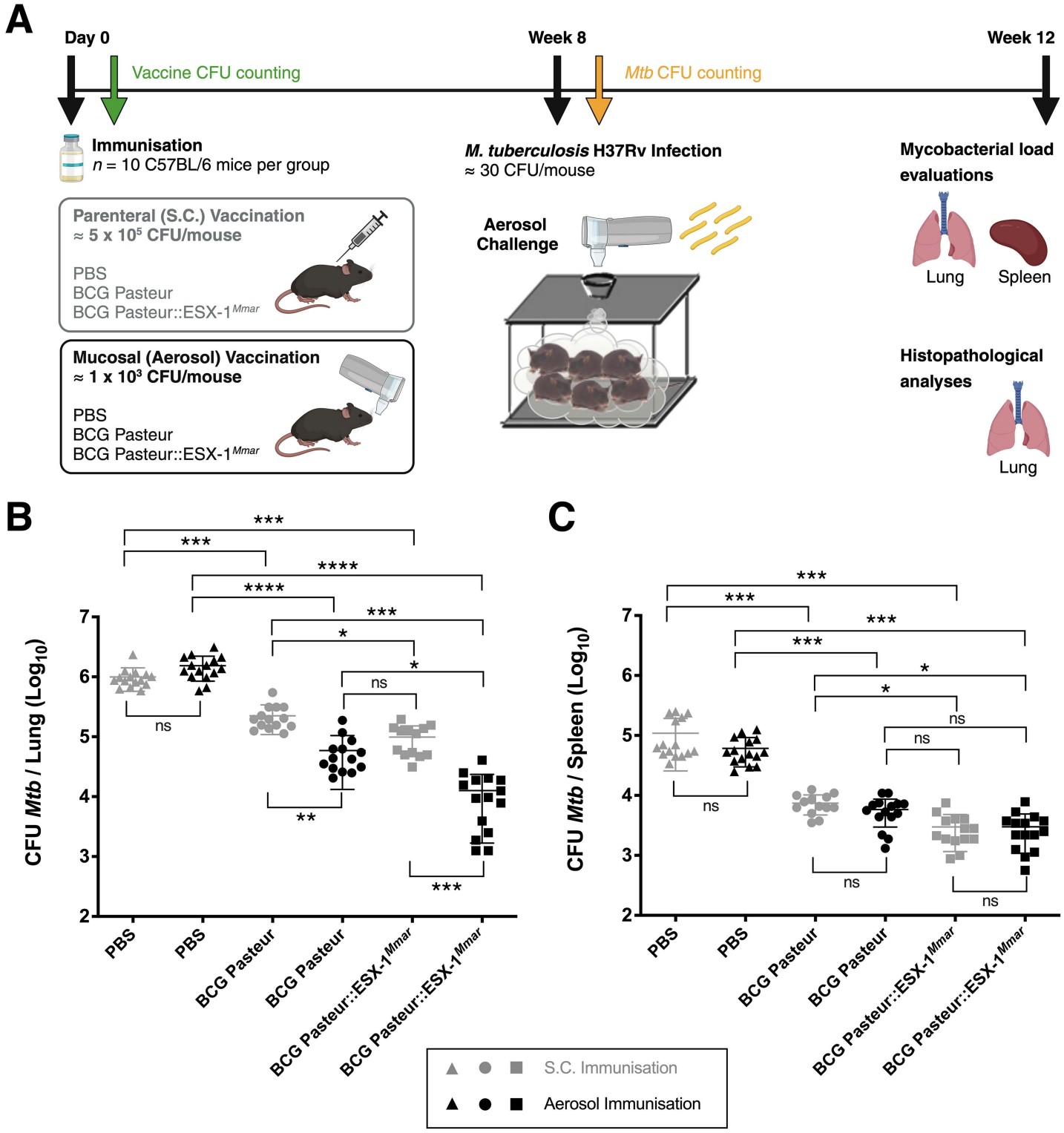

**Fig 7. Comparative evaluation of protection efficacy of different BCG vaccines in mice immunised via subcutaneous or aerosol routes against *M. tuberculosis* challenge.** A. Scheme to evaluate vaccine protection efficacy against infection with low dose of *M. tuberculosis* H37Rv strain via aerosol route. C57BL/6J mice (*n* = 10 per group, compiling data of two independent experiments) either left untreated or immunised subcutaneously with 5 x 10⁵ CFU/mouse or via aerosol route with ≈ 1 x 10³ CFU/mouse of BCG Pasteur or BCG::ESX-1*Mmar* vaccine. Four weeks post-infection, the

mycobacterial loads in the lungs and spleens of individual mice were quantified by CFU counting and lung histopathological evaluations were performed. Scheme was generated and licensed by BioRender under the agreement number KE294KWURI. Created in BioRender. Sayes, F. (2026) https://BioRender.com/j3opypx. B-C. C57BL/6J mice were immunised with BCG Pasteur or BCG::ESX-1$^{Mmar}$ vaccine strains via subcutaneous or aerosol routes. Mice were infected with ≈ 30 CFU/mouse of *M. tuberculosis* H37RV WT strain via aerosol route at eight weeks post-immunisation. The mycobacterial loads in the lung (B) and spleen (C) of individual mice were determined by CFU counting at four weeks post-infection. NS = not significant, *, **, ***, **** = statistically significant with $p < 0.05$, $p < 0.005$, $p < 0.001$ and $p < 0.0001$, respectively, as determined by Brown-Forsythe & Welch ANOVA tests for Multiple Comparisons with Dunnett's T3 Corrections. The figures and were elaborated by using Prism software. The results are accumulation of two independent experiments.

*M. tuberculosis* H37Rv WT strain used for mouse infection experiments was prepared from titrated frozen stocks kept at -80°C that were originally prepared from growing cultures in Dubos broth complemented with ADC at 37°C. Mycobacterial suspensions were washed twice with PBS and left to rest for 20 minutes before being collected as single-cell suspension. CFU were counted on Middlebrook 7H11 solid Agar medium complemented with OADC after 3–4 weeks of incubation at 37°C.

## Vaccination of mice for immunological studies and TB protection evaluations, organ preparations and CFU counting

Six- to seven-week-old female C57BL/6J mice (Janvier) were immunised subcutaneously with ≈ 5 x 10$^5$ CFU/mouse of different BCG vaccine strains (100 μl of volume given at the base of the tail) or via aerosol route generated from suspensions containing 2.5 x 10$^7$ CFU/ml of live-attenuated vaccine strains in order to reach an inhaled dose of ≈ 1 x 10$^3$ CFU of vaccine per mouse lung. Aerosol vaccination route was generated from mycobacterial suspension using clinically proven and commercial nebulizer (Pari LC Sprint) attached to mice-containing isolation chamber in L3 protection animal facility. Mice were placed in custom-made individual tubes in order to maximize homogeneous dose distribution. The received dose in the lungs was evaluated by CFU counting of lung homogenates from 2-3 mice up to 24 hours post-immunisation. Organs from the different groups of mice were used for CFU counting and histological analyses and others were used for immunological evaluations at different time-points post-immunisation.

Mice were infected via aerosol route with *M. tuberculosis*, using the same aerosol system and device as used for aerosol vaccination with BCG Pasteur and BCG::ESX-1$^{Mmar}$ strains. A mycobacterial suspension containing 1 x 10$^6$ CFU/ml of *M. tuberculosis* H37Rv strain was used to deliver a low dose of infection (≈ 30 CFU/mouse), as evaluated by CFU counting in the lung up to 24 hours post-infection. Four weeks later, mice were culled and lungs and spleens harvested for CFU counting and histopathological analyses. Individual organs were homogenized using a MillMixer organ homogenizer (Qiagen) and serial 5-fold dilutions were plated on Middlebrook 7H11 solid Agar medium complemented with 10% OADC, and/or containing Ampicillin (50 μg/ml), PANTA mixture (Polymyxin B, Amphotericin B, Nalidixic acid, Trimethoprim and Azlocillin, Becton Dickinson) or no antibiotics. CFU were counted after 3–4 weeks of incubation at 37°C.

## Immune cell preparations, *ex vivo* staining and flow cytometry

Adaptive immune cells from the lungs and spleens of immunised mice were prepared as previously described [55,56]. Briefly, lungs were washed with PBS and digested by treatment with 400 U/ml type-IV collagenase and DNase-I (Roche) for 30 minutes at 37°C. Single-cell suspensions were prepared by use of a GentleMacs (Miltenyi) and by passage through 100-μm diameter filters (Cell Strainer, BD Falcon). Lung and spleen cell suspensions were enriched in live lymphocytes T and B by using a density gradient centrifugation Ficoll medium (Lympholyte-M, Cedarlane) according to the manufacturer's protocol. The obtained lymphocyte layers were washed twice with PBS and incubated for 10 minutes at 10°C with FcγII/III receptor blocking anti-CD16/CD32 (BD 2.4G2 clone) mAb. Then, cells were incubated with appropriate dilutions of mAb containing different combinations of anti-CD3-APC-Cy7, anti-CD4-PerCP-Cy5.5, anti-CD8-Pacific Blue, anti-CD27-APC, anti-CD28-FITC, anti-CD44-PE, anti-CD45-BV510, anti-CD45RB-FITC, anti-CD62L-PE-Cy7, anti-CD69-FITC,

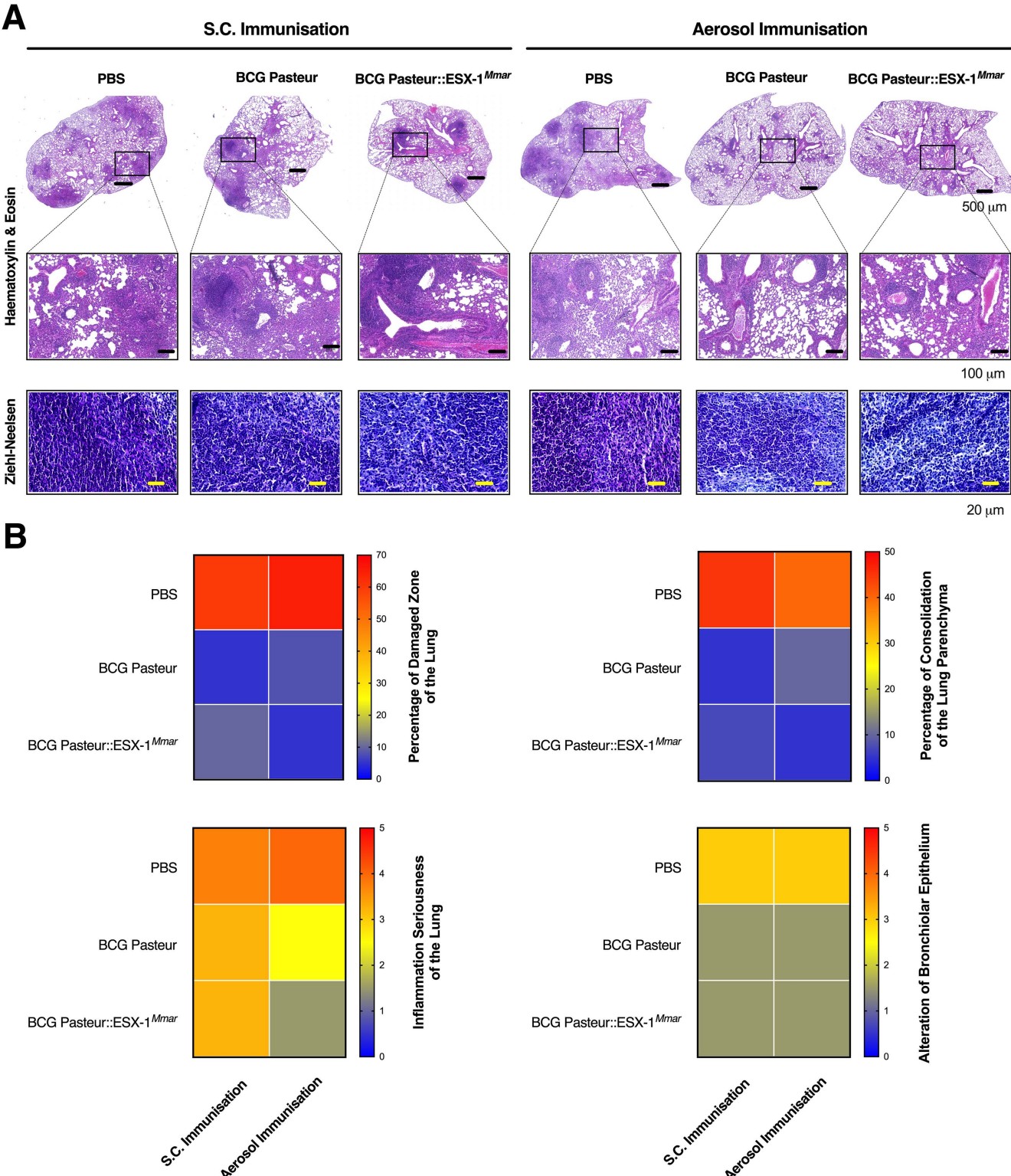

**Fig 8. Histological evaluations of the lungs of mice vaccinated with different BCG vaccines via subcutaneous or aerosol routes and infected with virulent *M. tuberculosis*.** A. Histopathological analyses on lung sections (left lobe) of *M. tuberculosis*-infected mice showing lesions and cell infiltrations, as evaluated by haematoxylin and eosin (H&E) staining method and acid-fast bacilli were observed by Ziehl-Neelsen staining method at four

weeks post-infection. Scale bars represent 500 µm (upper row), 100 µm (middle row) and 20 µm (lower row). B. Comparative lung histopathological evaluations and median scoring notes, based on different histological parameters, attributed to each group of mice. Five-scale severity grade was adopted for histopathological scoring; 1: minimal, 2: mild, 3: moderate, 4: marked and 5: severe. The figures were elaborated by using Zen and Prism softwares. The results are representative of two independent experiments.

anti-CD95 (Fas)-APC, anti-CD103-APC, anti-CD183 (CXCR3)-BV510, anti-CD196 (CCR6)-PE-Cy7 and anti-CD197 (CCR7)-PE-Cy7, prepared in FACS buffer (PBS containing 3% fetal bovine serum and 0.1% NaN3) during 30 minutes at 10°C sheltered from light. The stained cells were washed twice with FACS buffer and then fixed with an appropriate volume of 4% paraformaldehyde overnight at 10°C prior to sample acquisition. At least 500,000 events per sample, depending on experiment, were acquired by using LSR Fortessa flow cytometer system and BD FACSDiva software (BD Bioscience). The obtained data were analysed using FlowJo software (Treestar) and graphs were performed by Graph-Pad Prism software.

### T-cell stimulation and intracellular cytokine staining assay

Single-cell suspensions from spleens of immunised mice were obtained by tissue dissociation, homogenization and passage through 100 µm-pore filter. Pooled cells ($n = 3$ mice/group) were cultured at 7.5 x $10^6$ cells/well in flat-bottom 12-well plates (TPP) in the presence of 1 µg/ml anti-CD28 (clone 37.51) and 1 µg/ml of anti-CD49d (clone 9C10-MFR4.B) mAbs (BD Pharmingen) and stimulated with 10 µg/ml of Purified Protein Derivative (PPD) (Creative Diagnostics) or 4 µg/ml of recombinant ESAT-6 (Abcam) and CFP-10 (RayBiotech) proteins during 24 hours at 37°C et 5% $CO_2$ in RPMI 1640 medium GlutaMAX (Gibco), complemented with 10% fetal bovine serum (Gibco) followed by 4–5 hours of incubation with protein transport inhibitor containing Brefeldin A (Golgi Plug, BD Pharmingen), according to the manufacturer's instructions.

Cells were then harvested, washed twice with PBS and stained with Live/Dead-violet dye (Thermo Fisher) for 45 minutes at 10°C and sheltered from light. Cells were washed with FACS buffer and incubated with FcγII/III receptor blocking anti-CD16/CD32 (BD 2.4G2 clone) mAb for 10 minutes at 10°C. Cells were then incubated for 30 minutes with appropriate dilutions of anti-CD3ε-APC-eFluor780, anti-CD4-BUV737 and anti-CD8α-Pacific Blue or anti-CD8α-BV510 mAbs (BD Pharmingen) at 10°C and sheltered from light. The stained cells were washed twice with FACS buffer, permeabilized by use of Cytofix/Cytoperm kit (BD Pharmingen). Cells were then washed twice with PermWash buffer (BD Pharmingen) and incubated with appropriate dilutions of PerCP-Cyanine5.5-anti-IL-2 (clone JES6-5H4, eBioscience), PE-anti-TNF-α (clone 554419, BD Pharmingen), Alexa Fluor647-anti-IFN-γ (clone XMG1.2, eBioscience) and FITC-anti-IL-17A (clone TC11-18H10.1) mAbs during 30 minutes at 10°C and sheltered from light. Appropriate staining with control Ig isotypes was performed in parallel. Cells were subsequently washed with PermWash buffer and then with FACS buffer before fixation with 4% paraformaldehyde. At least 1,000,000 events per sample, were acquired by using LSR Fortessa flow cytometer system and BD FACSDiva software (BD Bioscience). The obtained data were analysed using FlowJo software (Treestar) and graphs were performed by GraphPad Prism software.

### Sera preparation and bronchoalveolar lavage (BAL) fluid collection

The blood samples were collected from the vaccinated mice under anaesthesia (100–200 µl per mouse) without any anti-coagulant and leaved in tube in a standing position for 30–45 minutes at room temperature. The tubes were centrifuged at 4000 rpm for 5 minutes at 10°C and the separated serum was collected, filtered and diluted in PBS (d 1/40) for fresh Ig quantifications by ELISA or stored at -80°C for long preservation.

For BAL fluid collection, mice were euthanised and their trachea were exposed under sterile conditions and opened carefully with a small incision to avoid blood contamination. A cannula (BD Insyte) connected to 1 ml syringe (BD Plastipak) containing 400–500 µl of previously cooled PBS was introduced slowly to the trachea and recovered back as

described by [57]. A volume of 300–400 µl of BAL fluid per mouse was obtained and maintained in cold conditions. The obtained BAL samples were centrifuged at 1800 rpm for 6 minutes at 10°C and the supernatant was collected. The BAL fluid was diluted in PBS (d 1/8) for fresh Ig quantifications by ELISA or stored at -80°C for long preservation.

### T-cell assay and ELISA

Pool of splenocytes and total lung homogenates of immunised mice ($n = 2$–3 per group) were cultured in flat-bottom 24-well plates (TPP) at $5 \times 10^6$ cells per well in RPMI 1640 medium GlutaMAX (Gibco), complemented with 10% fetal bovine serum (Gibco), 1% of MEM non-essential amino acids (Invitrogen, Life Technologies), $5 \times 10^{-5}$ M β-mercaptoethanol (Invitrogen, Life Technologies), 100 U/ml penicillin and 100 µg/ml streptomycin (Sigma-Aldrich) in the presence of 5 µg/ml of PPD or 2 µg/ml of ESAT-6 and CFP-10 proteins during 72 hours at 37°C and 5% $CO_2$. Medium alone and 4 µg/ml of rMalE protein were used as negative controls.

Cytokine productions in culture supernatants were quantified by ELISA. Nunc 96-well Maxisorp plates (Thermo-Fisher) were used and mAbs specific to IFN-γ (clone AN-18 for coating and clone R4-6A2 for detection) or IL-17A (clone TC11-18H10 for coating and clone TC11-8H4.1 for detection) were from BD Pharmingen.

Pro-inflammatory innate cytokine responses were *ex vivo* quantified in the lungs of vaccinated mice at different time-points. Individual organs ($n = 2$ per group) were homogenized using a MillMixer organ homogenizer. Lung suspensions were made in 1 ml PBS and filtered by passage through 0.45 µm low protein binding filters (Millex-HV). DuoSet ELISA kits (R&D Systems) were used for mouse IL-1β/IL-1F2 (DY401–05), IL-6 (DY406–05) and TNF-α (DY410–05) quantifications, in duplicate, in 100 µl volume per sample using Nunc 96-well Maxisorp plates and the recommended buffers according to the manufacturer's instructions.

Total and PPD-specific Immunoglobulin IgG and IgA responses were quantified in the sera and BAL fluids of vaccinated mice by ELISA. Nunc 96-well Maxisorp plates were coated with 5 µg/ml of PPD and over-night incubated at 10°C. HRP goat anti-mouse IgG (BioLegend) and HRP goat anti-mouse IgA (SouthernBiotech) were used for immunoglobulin quantification in diluted BAL fluids. Optical density was measured at 450 nm by FLUOstar Omega microplate reader (BMG LABTECH). Graphs were elaborated using GraphPad Prism software.

### Histology, immunohistochemical studies and histopathological scoring

Samples of respective lung lobes from immunised mice were fixed in 10% neutral buffered formalin for 72 hours, transferred to 70% ethanol and prepared for histopathological evaluations and scoring. Organs were embedded in paraffin for 4 µm microtome sectioning and stained with haematoxylin and eosin (H&E) and/or acid-fast bacilli Ziehl-Neelsen staining methods according to standard procedures. Monoclonal antibodies specific to F4/80 (Cell Signalling, D2S9R), Ly6B.2 (bio rad, MCA771G), B220 (BD Pharmingen, 550286), CD4 (Cell Signalling, D7D2Z), CD8 (Cell Signalling, D4W2Z) surface markers were used for immunohistochemical (IHC) studies in order to identify different immune cell populations in the lungs of mice.

Microscopic changes were qualitatively described and when applicable, scored semi-quantitatively, by histopathologists blinded to the conditions, using distribution qualifiers (i.e., focal, multifocal, locally extensive or diffuse), and a five-scale severity grade, i.e., 1: minimal, 2: mild, 3: moderate, 4: marked and 5: severe.

Analyses were performed at the Histology Platform of the Institut Pasteur and stained slides were evaluated with ZEISS Axio Scan.Z1 Digital Slide Scanner and Zen software version 3.9 (Zeiss).

### Statistical analyses

Graphs and statistical analyses were performed and analysed using GraphPad Prism software. The Unpaired *t* test with Welch's correction and Brown-Forsythe & Welch ANOVA tests for multiple comparisons with Dunnett's T3 Corrections were employed depending on the data to determine the significance values between the different groups.

PLOS Pathogens

## Supporting information

**S1 Fig. Animal well-being and histological evaluations of the lungs of mice immunised with different BCG vaccines via subcutaneous or aerosol routes. A.** Body-weight measurement and clinical symptom follow-up evaluations of C57BL/6J mice ($n = 4$ per group) that were either left untreated, immunised subcutaneously with $5 \times 10^5$ CFU/mouse or vaccinated via the aerosol route with ≈ $1 \times 10^3$ CFU/mouse of BCG Pasteur or BCG::ESX-1$^{Mmar}$ vaccine. Three-scale severity grades were adopted for mouse clinical scoring; 1: low, 2: moderate, 3: severe. **B.** Histological analyses on lung sections of individual vaccinated mice, evaluated by hematoxylin and eosin (H&E) staining method at eight weeks post-immunisation. Scale bars represent 500 μm (upper row) and 100 μm (lower row). **C.** Immunohistochemistry evaluations on lung sections of vaccinated mice detecting macrophage and neutrophil cell infiltrations at eight weeks post-immunisation. Error bars represent SD. NS = not significant, * and ** = statistically significant with $p < 0.05$ and $p < 0.005$, respectively, as determined by Unpaired $t$ test with Welch's correction. The figures were elaborated by using Zen and Prism softwares. The results are representative of two independent experiments.
(TIFF)

**S2 Fig. Immunogenicity of different BCG vaccines in the lungs of mice immunised via subcutaneous or aerosol routes. A.** Total numbers of CD3$^+$ CD4$^+$ and CD3$^+$ CD8$^+$ T cells recovered from the lungs of immunised mice at eight weeks post-immunisation, as determined as total numbers of cells in the Ficoll-treated lung fractions, multiplied by the percentages of CD4$^+$ or CD8$^+$ cells, respectively, as assessed by flow cytometry. **B.** T-cell IFN-γ responses in the lungs of BCG Pasteur or BCG::ESX-1$^{Mmar}$-vaccinated C57BL/6J mice via subcutaneous or aerosol routes, as assessed at eight weeks post-immunisation. Total lung cells of the immunised mice were pooled ($n = 2$ per group) and stimulated *in vitro* with PPD or a mixture of ESAT-6 and CFP-10 proteins during 72 hours at 37°C and 5% $CO_2$. Medium alone and MalE protein were used as negative controls. The IFN-γ was quantified in the culture supernatant by ELISA. Error bars represent SD. NS = not significant, *, **, *** and **** = statistically significant with $p < 0.05$, $p < 0.005$, $p < 0.001$ and $p < 0.0001$, respectively, as determined by Unpaired $t$ test with Welch's correction. The figures were elaborated by using Prism software.
(TIFF)

**S3 Fig. Gating strategy and cytometric analyses adopted to identify different subsets of Th1 cytokine-producing T effectors in the organ of vaccinated mice.** Pool of lung homogenates of BCG::ESX-1$^{Mmar}$-immunised C57BL/6J mice ($n = 4$ per group) stimulated *in vitro* with PPD or with a mixture of ESAT-6 and CFP-10 proteins for 24 hours at 37°C and 5% $CO_2$, or without antigen stimulation prior to Live/Dead, surface and intracellular cytokine staining procedures. PPD-stimulated cells were stained with Ig isotypes for quality and specificity controls. At least 1,000,000 events per sample were acquired for flow cytometric analyses. Cytometric plots represent 5% contours with outliers. The obtained data were analysed using FlowJo software and figures were elaborated by using Prism software.
(TIFF)

**S4 Fig. Th1 cytokine-producing CD4$^+$ and CD8$^+$ T effectors in the lungs of mice immunised with different BCG vaccines via subcutaneous or aerosol routes. A-B.** Frequency of lung CD4$^+$ and CD8$^+$ Th1 cytokine producing cells, as evaluated by *ex vivo* intracellular cytokine staining assay on live lung CD3$^+$ T lymphocytes recovered from vaccinated mice at two months post-immunisation without antigen stimulation (**A**). **B.** Composition of T effector subsets expressing IL-2, TNF-α and IFN-γ cytokines, represented as pie charts (**B**). **C.** Frequency of lung CD4$^+$ and CD8$^+$ Th1 cells from unvaccinated groups stimulated with PPD or a mixture of ESAT-6 and CFP-10 proteins for 24 hours at 37°C and 5% $CO_2$. Error bars represent SD. NS = not significant, *, **, *** and **** = statistically significant with $p < 0.05$, $p < 0.005$, $p < 0.001$ and $p < 0.0001$, respectively, as determined by Unpaired $t$ test with Welch's correction. At least 1,000,000 events per sample were acquired. The obtained data were analysed using FlowJo software and figures were elaborated by using Prism software.
(TIFF)

**S5 Fig. Profile of adaptive immune cells in the lungs of mice immunised with different BCG vaccines via subcutaneous or aerosol routes at eight weeks post-immunisation.** Expression of T-cell activation/recruitment markers on the surface of lung CD3+ CD4+ and CD3+ CD8+ T lymphocytes, recovered from BCG Pasteur- or BCG::ESX-1$^{Mmar}$-immunised C57BL/6J mice via subcutaneous or aerosol route, as evaluated by flow cytometric analyses at eight weeks post-immunisation. At least 500,000 events per sample were acquired. Cytometric plots represent 5% contours with outliers, representative of pool of two mice per group. The obtained data were analysed using FlowJo software and figures were elaborated by using Prism software. The results are representative of two independent experiments.
(TIFF)

**S6 Fig. Profile of adaptive immune cells in the lungs of mice immunised with different BCG vaccines via subcutaneous or aerosol routes at four weeks post-immunisation.** Expression of T-cell activation/recruitment markers on the surface of CD3+ CD4+ and CD3+ CD8+ T lymphocytes, recovered from BCG Pasteur- or BCG::ESX-1$^{Mmar}$-immunised C57BL/6J mice via subcutaneous or aerosol route, as evaluated by flow cytometric analyses at four weeks post-immunisation. At least 500,000 events per sample were acquired. Cytometric plots represent 5% contours with outliers, representative of pool of two mice per group. The obtained data were analysed using FlowJo software and figures were elaborated by using Prism software. The results are representative of two independent experiments.
(TIFF)

**S7 Fig. Induction of T effectors and T effector memory cell responses in the spleens of mice immunised with different BCG vaccines via subcutaneous or aerosol routes. A.** CD4+ and CD8+ T effectors expressing CXCR3 marker recovered from the spleens of the same C57BL/6J vaccinated mice in Fig 4 at eight weeks post-immunisation. Cell populations were gated on CD3+ CD28- CD44$^{high}$ CD95+ T lymphocytes. Cytometric plots represent 5% contours with outliers, representative of pool of two mice per group. **B.** Frequency of CD4+ and CD8+ T effector memory cells, defined as CD3+ CD44$^{high}$ CD62L- cells, among spleen CD4+ and CD8+ T lymphocyte populations, as evaluated by flow cytometric analyses at eight weeks post-immunisation. **C.** Mean Fluorescence Intensities (M.F.I.) of CD44 expression on total spleen CD3+ CD4+ and CD3+ CD8+ T cells recovered from the same vaccinated mice. Error bars represent SD. NS = not significant, *, ** and *** = statistically significant with $p < 0.05$, $p < 0.005$ and $p < 0.001$, respectively, as determined by Unpaired $t$ test with Welch's correction. At least 500,000 events per sample were acquired. T$_{EM}$: T effector memory cells. The obtained data were analysed using FlowJo software and figures were elaborated by using Prism software. The results are representative of two independent experiments.
(TIFF)

**S8 Fig. Induction of mycobacteria-specific long-lasting CD8+ T effectors and polyfunctional CD4+ Th1 responses in the spleens of mice immunised with different BCG vaccines via subcutaneous or aerosol routes.**
**A.** TNF-α- and IFN-γ-producing CD8+ T lymphocytes by intracellular cytokine staining assay applied on total splenocytes of mice at twelve weeks post-immunisation. Pool of splenocytes ($n = 2$ per group) were left untreated without antigen stimulation or stimulated *in vitro* with PPD for 24 hours at 37°C and 5% $CO_2$ prior to staining with cytokine-specific mAbs.
**B.** Unstimulated cells and PPD-stimulated cells were stained with Ig isotypes for quality and specificity controls. At least 500,000 events per sample were acquired for flow cytometric analyses. Cytometric plots are representative of pool of two mice per group. **C.** Frequency of Th1 cytokine-producing cells within spleen CD4+ T effectors recovered from vaccinated mice at twelve weeks post-immunisation. Total splenocytes of vaccinated mice ($n = 2$ per group) were stimulated *in vitro* with PPD for 24 hours at 37°C and 5% $CO_2$ prior to intracellular cytokine staining assay (IL-2, TNF-α and IFN-γ) and flow cytometric analyses. **D.** Total numbers of CD3+ CD4+ and CD3+ CD8+ T cells recovered from the spleens of vaccinated immunised mice at twelve weeks post-immunisation, as determined as total numbers of cells in the Ficoll-treated lung fractions, multiplied by the percentages of CD4+ or CD8+ cells, respectively, as assessed by flow cytometry. Error bars

represent SD. NS = not significant, as determined by Unpaired *t* test with Welch's correction. The obtained data were analysed using FlowJo software and figures were elaborated by using Prism software.
(TIFF)

**S1 Data. Raw data for the graphs contained in the manuscript.**
(ZIP)

## Acknowledgments

We thank Daria Bottai from the University of Pisa for fruitful discussions and advice. We also thank the members of the Institut Pasteur A3 animal facility, in particular Mathilde Dubot, Rachid Chennouf and Karim Sebastien for expert animal care. We are also grateful to the team of the Imaging platform of the Institut Pasteur, in particular Pierre-Henri Commere, for help and support.

## Author contributions

**Conceptualization:** Fadel Sayes, Roland Brosch.

**Data curation:** Fadel Sayes.

**Formal analysis:** Fadel Sayes, Wafa Frigui, Alexandre Pawlik, David Hardy, Roland Brosch.

**Funding acquisition:** Roland Brosch.

**Investigation:** Fadel Sayes, Wafa Frigui, Alexandre Pawlik, Cécile Tillier, Magali Tichit, David Hardy.

**Methodology:** Fadel Sayes, Wafa Frigui, Alexandre Pawlik, David Hardy, Roland Brosch.

**Resources:** Roland Brosch.

**Visualization:** Fadel Sayes, Cécile Tillier, Magali Tichit.

**Writing – original draft:** Fadel Sayes, Roland Brosch.

**Writing – review & editing:** Fadel Sayes, Roland Brosch.

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
