## [Decision Letter · Decision Letter 0]

26 Sep 2025

Improved Immune Responses and Tuberculosis Protection by Aerosol Vaccination with recombinant BCG expressing ESX-1 from Mycobacterium marinum.

PLOS Pathogens

Dear Dr. Brosch,

Thank you for submitting your manuscript to PLOS Pathogens. After careful consideration, we feel that it has merit but does not fully meet PLOS Pathogens's publication criteria as it currently stands. Specifically, all of the reviewers acknowledged the importance of the work, and the careful experimental approach.  Nonetheless, significant concerns remain. Therefore, we invite you to submit a revised version of the manuscript that addresses the points raised during the review process.

Please submit your revised manuscript within 60 days Nov 25 2025 11:59PM. If you will need more time than this to complete your revisions, please reply to this message or contact the journal office at plospathogens@plos.org. Please include the following items when submitting your revised manuscript:

We look forward to receiving your revised manuscript.

Kind regards,

David M. Lewinsohn

Academic Editor

PLOS Pathogens

Anne Jamet

Section Editor

PLOS Pathogens

Editor-in-Chief

PLOS Pathogens

Editor-in-Chief

PLOS Pathogens

orcid.org/0000-0002-7699-2064

**Journal Requirements:**

At this stage, the following Authors/Authors require contributions: Magalie Tichit, and Roland Brosch. Please ensure that the full contributions of each author are acknowledged in the "Add/Edit/Remove Authors" section of our submission form.

3) Some material included in your submission may be copyrighted. According to PLOSu2019s copyright policy, authors who use figures or other material (e.g., graphics, clipart, maps) from another author or copyright holder must demonstrate or obtain permission to publish this material under the Creative Commons Attribution 4.0 International (CC BY 4.0) License used by PLOS journals. Please closely review the details of PLOSu2019s copyright requirements here: PLOS Licenses and Copyright. If you need to request permissions from a copyright holder, you may use PLOS's Copyright Content Permission form.

Potential Copyright Issues:

- Figures: 1A and 6A.. Please confirm whether you drew the images / clip-art within the figure panels by hand. If you did not draw the images, please provide (a) a link to the source of the images or icons and their license / terms of use; or (b) written permission from the copyright holder to publish the images or icons under our CC BY 4.0 license. Alternatively, you may replace the images with open source alternatives. See these open source resources you may use to replace images / clip-art:

4) Please ensure that the funders and grant numbers match between the Financial Disclosure field and the Funding Information tab in your submission form. Note that the funders must be provided in the same order in both places as well.

**Reviewers' Comments:**

Reviewer's Responses to Questions

**Part I - Summary**

Reviewer #1: The manuscript by Sayes et al. characterizes the safety, immunogenicity, and protective efficacy of aerosol administration of live attenuated BCG and BCG-expressing the ESX-1 secretion system of Mycobacterium marinum (BCG::ESX-1Mmar). This vaccine has been previously characterized by the authors as a subcutaneous vaccine, and this study extends its evaluation to the aerosol route. The manuscript is well-written, and the experiments conducted are well executed and robustly support the conclusions drawn. Notably, while mucosal administration of BCG and other live attenuated vaccines have been extensively tested in mouse models, these studies typically employ the intranasal route for vaccine delivery. In contrast, the present study utilizes the aerosol route, which adds significant novelty to the work. This approach is also clinically relevant, as it aligns with recent clinical trials investigating BCG delivery via aerosol.

Reviewer #2: Tuberculosis remains a major cause of mortality worldwide, and current vaccines provide inadequate protection against adult pulmonary disease. BCG has notable limitations, including the absence of several immunodominant M. tuberculosisantigens and an impaired capacity to induce CD8⁺ T cell responses. To address these shortcomings, Sayes et al. generated a recombinant BCG strain expressing the Mycobacterium marinum ESX-1 system and compared it with conventional BCG when administered either parenterally or via aerosol.

The study has several strengths. The use of an aerosol route is highly relevant for TB, as it reflects the natural route of infection and reveals immune dynamics not captured by parenteral vaccination. The immunological analyses are comprehensive, including both humoral and cellular responses, and the protection data convincingly demonstrate that aerosol delivery of the recombinant strain enhances protection compared to standard BCG. The work is novel in combining a functional ESX-1 system from an orthologous species with an aerosol delivery platform, providing important insights into how to improve BCG-based vaccines.

Some limitations should also be acknowledged. The mechanistic link between lung-resident T cells and enhanced protection is not yet fully established. Polyfunctionality was assessed only in splenic T cells, which showed no significant differences between groups and are therefore unlikely to explain the increased protection observed with aerosolised BCG::ESX1^mar. A more direct evaluation of cytokine production by lung-derived CD4⁺ and CD8⁺ T cells, including both Th1 cytokines and IL-17, would substantially strengthen the conclusions and give insights into a possible mechanism. In addition, the immunophenotyping strategy and terminology could be refined. Since antigen is still present in the lung at the timepoints analyzed, these cells cannot yet be considered bona fide tissue resident memory T cells. Describing them instead as T cells expressing tissue-residency markers would provide greater accuracy.

Overall, the manuscript is clearly written, the experiments are well planned and executed, and the data presentation is of high quality. The study makes a novel and significant contribution to the TB vaccine field by highlighting how recombinant BCG strains and aerosol delivery can be leveraged to enhance protective immunity.

Reviewer #3: This study by Sayes et al. examines the protective potential of a recombinant BCG vaccine engineered to express a fully functional ESX-1 secretion system derived from the heterologous strain Mycobacterium marinum (BCG::ESX-1Mmar)

The BCG strain, currently used as a tuberculosis vaccine, has limited efficacy due to the partial loss of key antigens such as ESAT-6/CFP-10 expressed by the ESX-1 system, which is essential for phagosome rupture and cytosolic pathogen recognition. By reintegrating the genetic region encoding ESX-1 from M. marinum, a species closely related to M. tuberculosis, the team led by Professor Brosch at the Institut Pasteur created a version of BCG capable of more effectively stimulating both innate and adaptive immune responses. This modification allows BCG::ESX-1Mmar to activate the cGAS/STING/TBK1/IRF-3 pathway, leading to type I interferon production and activation of AIM2 and NLRP3 inflammasomes. This induces a robust immune response, with an increase in polyfunctional antigen-specific CD4+ Th1 T cells and effector CD8+ T cells against antigens shared with BCG, which was published the Brosch team (Gröschel et al. Cell Reports, 2017). Mouse model studies have demonstrated that subcutaneous administration of BCG::ESX-1Mmar offers enhanced protection relative to standard BCG against virulent M. tuberculosis strains, significantly lowering bacterial loads in the lungs and spleen of vaccinated animals. Additionally, this recombinant strain exhibits lower virulence compared to other BCG recombinants expressing ESX-1 from M. tuberculosis.

Here, the same team further evaluated the potency of BCG::ESX-1Mmar using aerosol vaccination. This route proved highly immunogenic, inducing both humoral and cellular responses in the airways. Compared with subcutaneous delivery, aerosol immunization generated higher frequencies of activated CD4⁺ and CD8⁺ T lymphocytes and stronger effector memory (TEM) responses in the lungs. Importantly, only aerosol vaccination elicited substantial tissue-resident memory T-cell (TRM) responses in the airways. It also induced potent polyfunctional Th1 effectors in the spleen and systemic IgG responses comparable to subcutaneous immunization. Finally, aerosol vaccination conferred superior TB protection and reduced lung pathology compared with subcutaneous injection, underscoring the relevance of investigating alternative routes of administration.

Taken together, this is a comprehensive study that addresses both general and TB antigen–specific immune responses. It represents high-quality work of significant importance—congratulations to the authors.

**Part II – Major Issues: Key Experiments Required for Acceptance**

Reviewer #1: The study contains all the required experiments to characterize in-depth the aerosol vaccines, and therefore I do not find that more key experiments are required.

Reviewer #2: In Figure 5, the authors assess the polyfunctionality of T cells derived from the spleen. While this is informative, it is not the most relevant compartment for the protective immune response in this model. The manuscript already provides compelling data on T cells expressing tissue residency markers, and evidence of cytokine production in lung cells (IFN-γ and IL-17 measured in supernatants by ELISA). However, the key missing experiment is a direct assessment of polyfunctionality of lung-derived CD4⁺ and CD8⁺ T cells. This is essential, since the splenic analysis failed to demonstrate significant group differences, strongly suggesting that splenic T cells are not driving the enhanced protection observed in the aerosolised BCG::ESX1mmar. Instead, the critical population likely resides in the lung, where local T cells would be expected to mount the polyfunctional Th1/Th17 responses responsible for protection. To validate this conclusion, the authors should perform intracellular cytokine staining of lung-derived lymphocytes, assessing Th1 cytokines (e.g., IFN-γ, TNF, IL-2) alongside IL-17. The authors should also revise their restimulation strategy. Restimulation with purified protein derivative (PPD) as well as with ESAT6–CFP10 would allow the authors to discriminate between immune responses driven by the parental BCG strain and those attributable to the recombinant antigen. If cell numbers are a limiting factor, simultaneous restimulation with both PPD and ESAT6/CFP10 antigens would also be acceptable. This distinction is important to determine whether the recombinant construct provides an additive effect beyond BCG alone. Finally, the PBS-vaccinated control group appears to be missing from the splenic ICS polyfunctionality data already presented. Inclusion of appropriate PBS controls in the proposed lung ICS experiments is important to establish background staining levels and to confirm that the observed cytokine production and polyfunctionality are truly vaccine-specific and not artifacts.

Reviewer #3: There are multiple routes of administration that can be used to target the lungs, including mucosal vaccination via intranasal droplets, intratracheal instillation, and nasal or aerosol delivery. To what extent can these be considered the same route of administration, and do they recruit the same T-cell populations?

How do the authors explain the differences in virulence observed between the ESX1 system from M. marinum and that from M. tuberculosis?

How does mucosal vaccination compare with the prime–boost strategy reported by the same group (Lopez et al.)?

A major study published by Darrah et al. in Nature (January 2020) demonstrated enhanced vaccine protection following intravenous administration. How do the authors reconcile their findings with those results?

**Part III – Minor Issues: Editorial and Data Presentation Modifications**

Reviewer #1: 1. The data on IL-17 production in the lungs are presented in the supplementary material. Given the critical role of this cytokine in the protective efficacy of mucosal vaccines, I recommend moving these results to the main figures. Additionally, the authors currently show only PPD-specific IL-17 production. I suggest including data on ESAT-6/CFP-10-specific IL-17 production, as this could provide insight into the enhanced protection observed with aerosol BCG::ESX-1Mmar compared to BCG.

2. Regarding the vaccine dose used for aerosol administration, previous studies have indicated that protection mediated by intratracheal BCG (but not subcutaneous BCG) is dose-dependent (DOI: 10.1128/CVI.00700-13). Have the authors considered testing higher doses of the aerosol vaccine? Alternatively, are there physical limitations in the model that prevent nebulization of higher bacterial loads?

3. The authors propose that aerosol vaccination is well-suited for developing vaccine strategies targeting adolescents and adults, a current priority in the field. In this context, do the authors have data on the protective efficacy of aerosol vaccination in mice previously primed with subcutaneous BCG? Such data could better mimic the immunological context of adolescents and adults in TB-endemic regions.

Reviewer #2: Line 65: Please delete the word “been” for clarity.

Supplementary data: There appear to be mismatches between supplementary figures, their references in the main text, and their legends. Please cross-check and correct.

Line 419: The phrase “…used in protection studies” is unclear. Does this mean that different cohorts of mice were used for immunogenicity versus protection experiments? Please clarify.

Line 431: Was the aerosol system used to infect mice with M. tuberculosis the same as that used for aerosol vaccination? Please specify in the text.

Line 447: After preparing single-cell suspensions and washing with PBS, was an Fc receptor block and live/dead cell staining step performed prior to antibody staining? This should be clarified in the Methods.

Line 471: Similarly, please clarify whether live/dead staining was included in the intracellular cytokine staining (ICS) assay following T cell stimulation.

IgG/IgA reporting: Antibody responses are presented as absorbance values but described in the text as “titres.” By definition, titres refer to the reciprocal dilution at which a sample meets the threshold (e.g., mean of negative controls). If possible, titres should be calculated, as they are more informative. If only absorbance is reported, the authors should specify the dilution of serum or BAL at which the absorbance was measured.

Methods: Please include a description of how bronchoalveolar lavage (BAL) was collected.

Figure presentation: IL-17 data are currently presented in the supplementary material. Given their relevance, they should be moved into the main figures.

Definition of T cell subsets: The terminology used to define tissue-resident memory T cells (TRMs) and effector/effector memory T cells (TEM) requires clarification. TRMs are typically defined in the absence of ongoing antigen stimulation. Since BCG is still present in the lung at the analyzed timepoints, the cells described cannot be classified as bona fide TRMs. It would be more accurate to state that they express tissue-resident markers. Similarly, in the presence of antigen, TEMs would more accurately be termed effector T cells. Furthermore, TEMs in mice are commonly defined as CD44⁺CD62L⁻, whereas the authors use CD44⁺CD28⁻CD95⁺. Please provide justification in the Results for this gating strategy and cite relevant references.

Reviewer #3: Figure 1.

• Panel A: The display of mice could be improved by applying a consistent color code (e.g., ranging from white to grey to black) to represent each type of post-mortem analysis, which would facilitate the reader’s understanding of the workflow.

• Panel B: I recommend maintaining the same color scheme as in panel A to ensure visual consistency across the figure.

• Panel C: It would be helpful to include Lung IL-6 and Lung TNF-α on the Y-axis.

The experiments involve data from a remarkably large number of animals. Could the number of animals used be reduced without compromising statistical power? Additionally, how do the authors account for the considerable variability observed in the results?

PLOS authors have the option to publish the peer review history of their article (what does this mean? ). If published, this will include your full peer review and any attached files.

**Do you want your identity to be public for this peer review?** For information about this choice, including consent withdrawal, please see our Privacy Policy .

Reviewer #1: No

Reviewer #2: **Yes:** Claudio Counoupas

Reviewer #3: No

**Figure resubmission:**

**Reproducibility:**



---

## [Decision Letter · Decision Letter 1]

12 Feb 2026

Dear Pr. Brosch,

We are pleased to inform you that your manuscript 'Improved Immune Responses and Tuberculosis Protection by Aerosol Vaccination with recombinant BCG expressing ESX-1 from Mycobacterium marinum.' has been provisionally accepted for publication in PLOS Pathogens.

Best regards,

David M. Lewinsohn

Academic Editor

PLOS Pathogens

Anne Jamet

Section Editor

PLOS Pathogens

Sumita Bhaduri-McIntosh

Editor-in-Chief

PLOS Pathogens

orcid.org/0000-0003-2946-9497

Michael Malim

Editor-in-Chief

PLOS Pathogens

orcid.org/0000-0002-7699-2064

Reviewer Comments (if any, and for reference):

Reviewer's Responses to Questions

**Part I - Summary**

Reviewer #2: Tuberculosis remains a leading cause of mortality worldwide, and current vaccines provide inadequate protection against adult pulmonary disease. One of the major limitations of BCG is its inability to induce robust CD8⁺ T cell responses, largely due to the absence of key immunodominant antigens and a non-functional ESX-1 secretion system. To overcome this, Sayes et al. developed a recombinant BCG strain expressing the Mycobacterium marinum ESX-1 system and evaluated its immunogenicity and protective efficacy when delivered either parenterally or via the aerosol route.

In this revised version of the manuscript, the authors have comprehensively addressed all points raised by the three reviewers. The revisions have substantially strengthened the study, improving clarity, experimental rigor, and presentation/interpretation of the data. In particular, the authors have refined their immunological analyses, clarified methods and improved their data presentation. As a result, the manuscript is now significantly stronger than the original submission.

The study convincingly demonstrates that aerosol delivery, but not parenteral administration, enables lung seeding of the vaccine strain and induces robust local immune responses. Aerosol vaccination elicits strong IFN-γ and IL-17 responses in the lung, alongside measurable IgG and IgA antibody production. Detailed cellular analyses show that this route preferentially expands lung T cells expressing effector memory and tissue-residency markers in both the CD4⁺ and CD8⁺ compartments. Functional analyses further reveal that only the recombinant strain induces IFN-γ responses to ESAT6 and CFP10 restimulation, consistent with restoration of ESX-1–dependent antigenicity. Most importantly, the protection studies demonstrate markedly enhanced protection in mice vaccinated via the aerosol route with the BCG:ESX-1mmar, representing the key strength of the work.

Overall, this manuscript is well written, technically sound, and it places an important problem in the TB vaccine field regarding improving BCG and inducing mucosal immunity. The combination of a recombinant BCG platform with aerosol delivery is well justified and interesting for the field. With the authors having thoroughly addressed the reviewers’ comments, I recommend this manuscript for publication.

**Part II – Major Issues: Key Experiments Required for Acceptance**

Reviewer #2: The authors have satisfactorily addressed all major issues raised by the reviewers in the previous round of review. The additional data, clarifications, and revisions provided adequately validate the study’s conclusions and substantially strengthen the presentation and interpretation of the results. No further key experiments or major modifications are required at this stage.

**Part III – Minor Issues: Editorial and Data Presentation Modifications**

Reviewer #2: The authors have addressed all major points raised by the reviewers thoroughly and thoughtfully. As a result, the manuscript is greatly improved and now presents the data in a clearer, more impactful, and more coherent manner. The overall quality of the manuscript has been substantially increased.

Missing methodological details for the detection of IL-1β, IL-6, and TNF-α in the lungs of infected mice (Figure 1). Please specify the assay used and relevant experimental conditions in the Methods.

Figure 5: The legend states that Panel C shows both percentages and absolute numbers; however, only absolute numbers are presented in the figure. Please correct the legend or update the figure accordingly.

In the T cell assay and ELISA Methods sections, please specify what the ELISA plates were coated with and clearly define the antigen specificity of the mycobacteria-specific immunoglobulins being measured.

The following very minor comments are intended as suggestions to further enhance clarity and data presentation and I leave it to the authors whether to incorporate them.

For functional readouts such as cytokine expression (Figure 3C), I would recommend avoiding the use of MFI calculated across total CD4+ or CD8+ T cell populations. Such values are heavily influenced by cytokine-negative cells and therefore do not accurately reflect per-cell cytokine production. Instead, MFI should be calculated within cytokine-positive gates (for example IL-17+ or IFN-γ+ cells), alongside reporting of responder frequencies. Together, these metrics provide a more biologically meaningful assessment of response magnitude and intensity. From a statistical perspective, calculation of MFI within cytokine-positive populations in negative or unvaccinated control groups can be unreliable, as cytokine-positive cells are often absent or represent background or non-specific staining. Very low event numbers may therefore introduce high variability and potentially misleading representations. Alternatively, MFI could be reported only for vaccinated groups in which clear cytokine-positive populations are present, or integrated metrics such as gMFI could be used to account for both response frequency and intensity.

Similarly in Figure 4C the presentation of CD44 expression as mean fluorescence intensity (MFI) across total CD4⁺ and CD8⁺ T cell populations is of limited interpretative value, as it averages phenotypically distinct naïve (CD44^low) and activated or memory (CD44^high) subsets. This approach makes it difficult to determine whether observed differences arise from changes in the frequency of CD44^high cells, altered expression levels within activated cells, or a combination of both. A more informative analysis would include reporting the proportion of CD44^high cells and or CD44 MFI within defined activated or memory gates, which would better characterise vaccine-induced T cell activation.

PLOS authors have the option to publish the peer review history of their article (what does this mean? ). If published, this will include your full peer review and any attached files.

**Do you want your identity to be public for this peer review?** For information about this choice, including consent withdrawal, please see our Privacy Policy .

Reviewer #2: **Yes:** Claudio Counoupas

---

## [Editor Report · Acceptance letter]

Dear Pr. Brosch,

We are delighted to inform you that your manuscript, "Improved Immune Responses and Tuberculosis Protection by Aerosol Vaccination with recombinant BCG expressing ESX-1 from Mycobacterium marinum.," has been formally accepted for publication in PLOS Pathogens.

Best regards,

Sumita Bhaduri-McIntosh

Editor-in-Chief

PLOS Pathogens

orcid.org/0000-0003-2946-9497

Michael Malim

Editor-in-Chief

PLOS Pathogens

orcid.org/0000-0002-7699-2064